# Understanding the Skill Gap in Recurrent Language Models: The Role of the Gather-and-Aggregate Mechanism

**Aviv Bick** [1]  **Eric P. Xing** [1 2]  **Albert Gu** [1 3]

## Abstract

State-space models (SSMs) offer efficient alternatives to Transformers for long sequences, but their fixed-size recurrent state limits capability on algorithmic tasks, such as retrieving past context. In this work, we examine how in-context retrieval operates in Transformer- and SSM-based language models and find that both rely on a ***Gather-and-Aggregate (G&A)*** mechanism: a *Gather Head* extracts relevant information from context, which an *Aggregate Head* integrates into representation. In both architectures, G&A concentrates in a few heads, forming bottlenecks even for simple retrieval. For example, disabling a single Gather *or* Aggregate Head in a pruned Llama-3.1-8B impairs retrieving the correct answer letter in MMLU, reducing its accuracy from 66% to 25%. Moreover, this retrieval bottleneck can obscure knowledge demands of tasks as the pruned model succeeds on MMLU with functioning G&A heads yet fails on other knowledge benchmarks. The bottleneck similarly extends to tasks where SSMs typically underperform, like GSM8K, BBH, and dialogue. We show that SSMs' retrieval challenges manifest in these heads, creating smoother attention patterns instead of the sharp transitions effective G&A requires. Thus, ***the Transformer-SSM retrieval gap exists in just a few heads, rather than the entire language model***. This suggests a unified explanation for Transformer vs. SSM performance gap while showing how to merge their strengths. We find that pretrained hybrid models, where SSMs are combined with attention layers, delegate the role of Aggregate Heads to attention. Similarly, replacing a single G&A head in a pretrained SSM with an attention variant boosts retrieval and benchmark scores.

[1]Carnegie Mellon University, Pittsburgh, PA, USA [2]MBZUAI, Abu Dhabi, UAE [3]Cartesia.ai. Correspondence to: Aviv Bick <abick@cs.cmu.edu>.

*Proceedings of the $42^{nd}$ International Conference on Machine Learning*, Vancouver, Canada. PMLR 267, 2025. Copyright 2025 by the author(s).

## 1. Introduction

Transformers have driven major breakthroughs in language modeling, but their quadratic scaling with sequence length has renewed interest in recurrent alternatives like state-space models (SSMs). SSMs offer linear scaling with a fixed-size memory while remaining competitive across many tasks. However, replacing a large, explicit history cache with a compact state representation poses a fundamental tradeoff. SSM-based language models often struggle with the skill of "in-context retrieval"—the ability to precisely locate and extract tokens presented earlier in the sequence, such as recalling a phone number mentioned several paragraphs ago (Jelassi et al., 2024). This "algorithmic" challenge, which Transformers handle effectively with direct access to past tokens, has been identified as a key factor in the performance gap between Transformer models and SSM variants (Arora et al., 2023).

Previous work on SSM retrieval limitations has treated language models as black boxes or focused on simplified theoretical setups (Wen et al., 2024; Jelassi et al., 2024; Waleffe et al., 2024). Meanwhile, Transformer research shows that algorithmic capabilities, including retrieval, often concentrate in specific attention heads (Lieberum et al., 2023; Wu et al., 2024). This raises a key question: *Can the Transformer-SSM performance gap be traced to algorithmic capabilities in a few specific heads?* Clarifying this will enable a more targeted comparison of the two architectures, helping determine whether SSMs can serve as viable building blocks for language modeling or require fundamental modifications.

The MMLU benchmark (Hendrycks et al., 2021) exemplifies this performance gap. While SSMs match Transformers on many knowledge benchmarks, they significantly underperform on MMLU unless trained substantially longer (Waleffe et al., 2024). This gap stems from MMLU's evaluation format: rather than scoring the likelihood of full-text answers, models must select letter labels (A, B, C, or D) corresponding to the correct option. This creates a dual challenge requiring both *factual knowledge* and *retrieval* capabilities (Lieberum et al., 2023).

To isolate these distinct challenges, we show that both archi-

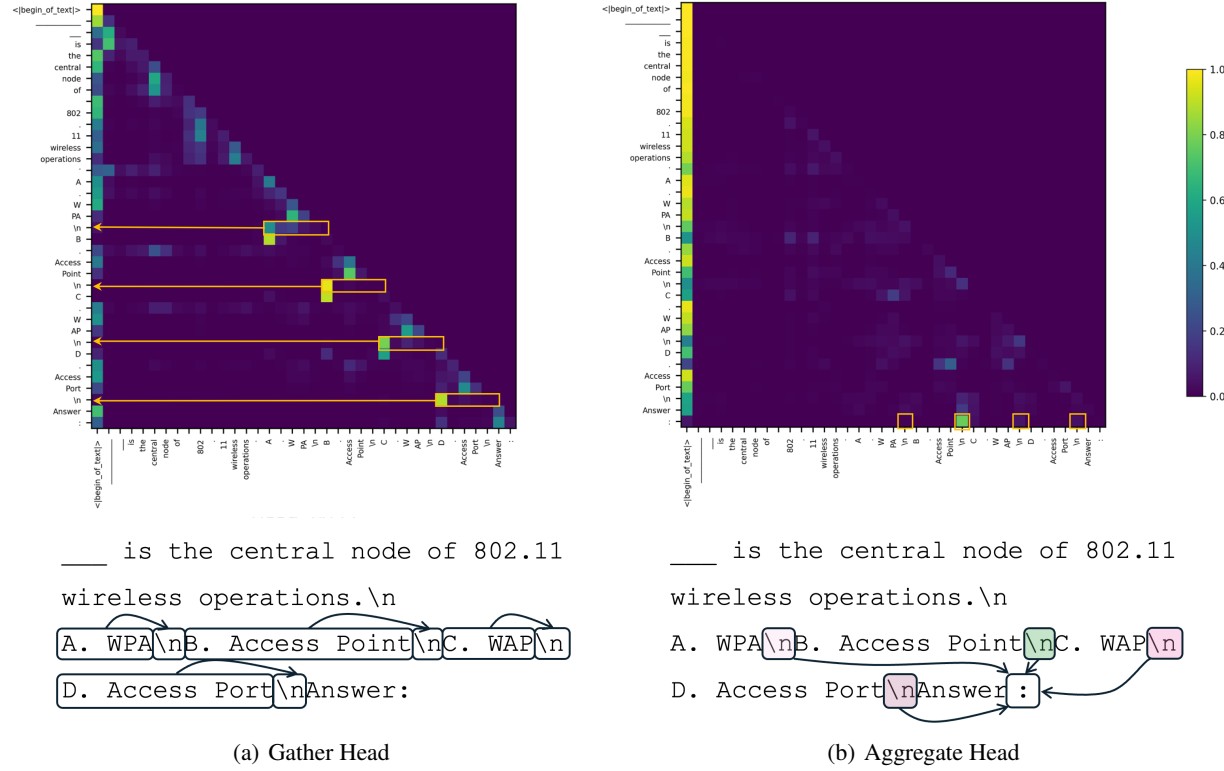

(a) Gather Head          (b) Aggregate Head

*Figure 1.* **An illustration of *Gather Head* (left matrix) at `L16H22` and *Aggregate Head* (right matrix) at `L17H24` of Llama-3.1-8B for a multiple-choice question.** The Gather Head identifies critical segments and summarizes each segment to its last token "\n" as shown in Figure 1(a), condensing key information into the residual stream. The Aggregate Head then combines these summaries as illustrated in Figure 1(b), giving a significantly higher score to the second "\n," which corresponds to the correct answer (B).

tectures have a clear division of labor: factual knowledge extracted across layers and heads, but only a small number of specialized heads retrieve the correct labels from input context. This becomes particularly evident in our experiments where a 4B-parameter pruned Llama-3.1-8B achieves a strong MMLU score of 66%, which plummets to 25% when disabling a single retrieval head. Moreover, these heads obscure the true knowledge demands of such tasks—despite strong MMLU performance, the pruned model performs poorly on knowledge-focused benchmarks. Similar dependence extends beyond MMLU to mathematical reasoning (GSM8K), multi-step logical reasoning (BBH), and dialogue formats (ARC-C Chat), demonstrating that retrieval capabilities, not knowledge deficits, drive SSM underperformance in these domains.

To understand how these specialized heads function, we investigate their underlying mechanism. We find that both Transformer- and SSM-based language models converge on a similar retrieval framework: a **Gather-and-Aggregate (G&A)** mechanism. Specifically, a *Gather Head* identifies and condenses segments of input tokens into a single vector representation (Figure 1(a)), which an *Aggregate Head* subsequently processes to extract the necessary information

(Figure 1(b)). This finding indicates that the two language models rely on similar retrieval strategies, despite architectural differences. Consequently, *their retrieval gap reduces to the effectiveness with which a few specific heads implement the G&A mechanism.*

Comparing G&A implementations, we find SSMs' known retrieval challenges manifest in how key heads implement G&A. SSMs exhibit smoother attention patterns, making sharp token shifts difficult and requiring more heads to achieve comparable performance. Moreover, we show how hybrid models (Glorioso et al., 2024; Lieber et al., 2024), which integrate a few attention layers into predominantly SSM-based architectures, overcome SSM's limitations. When trained from scratch, hybrid models naturally delegate Aggregate Head functionality to attention layers. Conversely, in pretrained SSM models, replacing a single G&A instance with an attention-based one yields a significantly greater MMLU improvement than any other attention placement.

Our experiments are publicly available.[1]

---

[1] https://github.com/goombalab/Gather-and-Aggregate

## 2. Related Work

**Sequence Models.** Transformer-based models with self-attention (Vaswani et al., 2023; Brown et al., 2020; Touvron et al., 2023) dominate language modeling for their scalability and performance. Recently, recurrent models like structured state-space models (SSMs) (Gu et al., 2022; Gu & Dao, 2023; Dao & Gu, 2024) gained interest for their sub-quadratic efficiency and competitive results on long sequences. Other recurrent models include (Dao et al., 2022; Sun et al., 2023; Katharopoulos et al., 2020; Yang et al., 2024a; Zhang et al., 2024b; Beck et al., 2024; Qin et al., 2024; Peng et al., 2024; Yang et al., 2024b). We use "SSM" to refer to these models as well, as our findings broadly apply.

Despite SSMs' efficiency, they struggle with tasks requiring precise token interactions, such as in-context retrieval. Their fixed-size hidden state limits memory capacity, making it difficult to retain detailed past information. To address these limitations, recent work (Lieber et al., 2024; Glorioso et al., 2024; Waleffe et al., 2024; Ren et al., 2024; Wang et al., 2024a; Paliotta et al., 2025) has adopted a hybrid approach, integrating attention layers with a majority of SSM layers, trading efficiency for improved capabilities such as retrieval.

Throughout this study, we examine four architectures: *Llama-3.1-8B-Instruct* (Touvron et al., 2023), a standard Transformer; *Falcon-Mamba-7B-Instruct* (Zuo et al., 2024), a Mamba-1 SSM without MLP layers; *Llamba-8B-Untied* (Bick et al., 2025), a Mamba-2 SSM distilled from Llama-3.1-8B using MOHAWK (Bick et al., 2024); and *Zamba-2-7B-Instruct* (Glorioso et al., 2024), a hybrid model with a 6:1 ratio of Mamba2 to attention layers and only two shared attention layers. For brevity, we refer to these as *Llama-3.1-8B*, *Falcon-Mamba-7B*, *Llamba-8B*, and *Zamba-2-7B*, respectively.

Central to understanding the performance differences between these architectures is *token mixing*: how models propagate and combine information across token positions. In self-attention, tokens selectively aggregate context via learned weights, while SSMs use structured recurrences with more restricted patterns. Ali et al. (2024); Dao & Gu (2024) connect SSM token mixing to attention's matrix-multiplication form, showing Mamba-1 mimics many low-dimensional attention heads. Mamba-2 (Dao & Gu, 2024; Bick et al., 2025; Waleffe et al., 2024) uses fewer, larger heads, increasing expressivity. These insights allow us to view all sequence models as structured matrices over time. Thus, we generalize token mixing beyond self-attention, referring to these mechanisms as *Temporal Mixing Heads* (or simply *Mixing Heads*).

**Retrieval Tradeoffs Between Transformers and SSMs.** SSMs rely on compressed, continuously updated hidden states, unlike Transformers, which cache full context. This structural difference impacts retrieval performance across benchmarks. Waleffe et al. (2024) found that Mamba-based models require extensive training to perform well on MMLU, while Wen et al. (2024) identified weaknesses in in-context retrieval tasks, such as associative recall. Jelassi et al. (2024) theoretically and empirically confirmed that SSMs have difficulty with precise copying. Arora et al. (2023) demonstrated that associative recall capabilities account for the majority of performance differences between attention-based and gated-convolution models. Park et al. (2024) shows that SSMs underperform on tasks requiring non-standard retrieval, and proposed a hybrid architecture to address this limitation. Arora et al. (2025) highlighted a recall-throughput tradeoff, with attention offering stronger recall at higher cost. Blouir et al. (2025) complemented this view by enhancing retrieval in SSMs through training improvements without architectural changes. Unlike previous work, we show these limitations originate from a small subset of retrieval heads rather than the entire model.

**Mechanistic Interpretability of Mixing Heads.** Most interpretability studies focus on attention heads in LLMs (Elhage et al., 2021; Lieberum et al., 2023; Zhang et al., 2024a; Olsson et al., 2022; Yu et al., 2024; Merullo et al., 2024; Rai et al., 2024; Zheng et al., 2024; Wu et al., 2024; Tulchinskii et al., 2024; Sanford et al., 2024), while relatively little attention has been given to recurrent mechanisms (Ali et al., 2024). Olsson et al. (2022) and Elhage et al. (2021) showed that induction heads emerge to support in-context learning. Subsequent work traced their training dynamics (Bietti et al., 2023), and mapped factual recall circuits (Meng et al., 2023).

Lieberum et al. (2023) identified two key heads in Chinchilla-70B—*Content Gatherer* and *Correct Letter*—that, with MLPs, support label prediction in multiple-choice tasks. Wu et al. (2024) found that fewer than 5% of heads, termed *Retrieval Heads*, are critical for retrieval, Chain-of-thought, and reasoning.

We refine the understanding of how heads support retrieval by showing that, in both Transformers and SSMs, retrieval arises from a coordinated mechanism—not a single head. One head identifies the target location, while another aggregates its content. This view extends prior work in three ways. First, the "Correct Letter" heads identified by Lieberum et al. (2023) exhibit broader retrieval behavior than previously reported, going beyond multiple-choice selection. Second, retrieval depends on coordinated interactions between heads, challenging the interpretation in Wu et al. (2024) that treated individual heads as independently responsible. Third, we demonstrate that this mechanism also arises in SSMs—despite their lack of attention—highlighting its architectural generality. We introduce the term Aggregate

Head to better reflect these empirical findings, while retaining "Gather" as shorthand for the "Content-Gatherer" heads described Lieberum et al. (2023).

Concurrent with our work, Wiegreffe et al. (2025) extended Lieberum et al. (2023)'s findings by analyzing training dynamics and cases of poor model performance. They introduced synthetic tasks to disentangle knowledge from multiple-choice formatting and identified mechanisms by which models adapt to different answer symbols. While their goals align with ours, we focus on the algorithmic aspects of in-context retrieval in multiple-choice settings across architectures.

## 3. A Case Study: Retrieval in MMLU

The MMLU benchmark (Hendrycks et al., 2021) spans 57 diverse tasks and is often viewed as a test of world *knowledge*. However, its multiple-choice format also places demands on *retrieval*: models must map question-answer context to the correct letter label (A–D), as illustrated below:

```
___ is the central node of 802.11
wireless operations.
A. WPA
B. Access Point
C. WAP
D. Access Port
Answer:
```

Despite performing well on many factual benchmarks, SSMs underperform on MMLU unless trained substantially longer (Waleffe et al., 2024). Given their known retrieval limitations (Jelassi et al., 2024), we ask whether this gap stems from a failure of knowledge or of retrieval. To investigate, we analyze three representative models: Llama-3.1-8B-Instruct (Transformer), Falcon-Mamba-7B-Instruct (Mamba-1), and Llamba-8B-Untied (Mamba-2). We compare their performance on MMLU versus benchmarks that emphasize factual reasoning with minimal reliance on in-context retrieval. For clarity, we omit instructional and embedding-tying suffixes from model names in our discussion.

Our investigation reveals two key findings:

1. Models can perform well on MMLU with limited knowledge, suggesting that the task primarily reflects in-context retrieval for the evaluated models. This adds to concerns raised by prior work (Wang et al., 2024b; Zhao et al., 2024).

2. Both Transformer- and SSM-based models encode retrieval abilities in a small set of specialized heads, extending prior work that focused solely on Transformers (Lieberum et al., 2023; Wu et al., 2024).

These insights, combined with earlier findings about perplex-

ity gaps between attention-based models and SSM variants (Arora et al., 2023), support our main claim: the general performance gap between Transformer- and SSM-based language models can be traced to specific algorithmic capabilities within a small subset of heads.

Our analysis proceeds bottom-up: we begin by contrasting MMLU with these knowledge-focused tasks (Section 3.1), then localize MMLU performance to a single critical layer (Section 3.2). We show coordination between two layers (Section 3.3), narrow the effect to a small set of heads (Section 3.4), and demonstrate that these implement a shared retrieval mechanism across models (Section 3.5).

### 3.1. Evaluation Benchmarks.

Throughout the paper, we evaluate performance on two categories of benchmarks: MMLU and what we term ***Knowledge-Focused Tasks***. MMLU's format distinctively presents multiple-choice questions, requiring models to process lengthy contexts, then identify the correct option by outputting the corresponding letter label (A, B, C, or D). While MMLU is widely used, studies have noted that strong performance may not reflect true understanding or knowledge (Alzahrani et al., 2024; Wang et al., 2024b; Zhao et al., 2024).

In contrast, Knowledge-Focused Tasks include ARC-Challenge and ARC-Easy (Clark et al., 2018), PIQA (Bisk et al., 2019), Winogrande (Sakaguchi et al., 2019), OpenBookQA (Mihaylov et al., 2018), and HellaSwag (Zellers et al., 2019). These benchmarks primarily assess factual knowledge and commonsense reasoning with minimal reliance on recent context retrieval. The core difference lies in how predictions are evaluated: MMLU requires selecting a letter label tied to the correct answer, while the other benchmarks score the likelihood of complete answer texts—shifting the focus from label retrieval to how well the model's internal knowledge supports each option.

### 3.2. A Crucial Layer for MMLU

Motivated by the struggle of SSMs with MMLU (Waleffe et al., 2024) and by the fact that Transformers use specialized components to solve multiple-choice questions (Lieberum et al., 2023), we examine whether solving this task utilizes different components in SSM-based language models as well. Since all language models utilize residual skip connections, eliminating a layer could reveal its specific contribution while keeping the rest of the information flowing.

We iteratively remove layers from the end of the model and evaluate performance on both MMLU and the knowledge-focused tasks. Appendix A.1 presents performance trends for Llama-3.1-8B as layers are progressively removed, and

Appendix A.1 extends this analysis to Falcon-Mamba-7B and Llamba-8B. For Llama-3.1-8B and Llamba-8B, we ablate both the mixer and its downstream MLP, since the latter likely depends on the former. As shown in Appendix A.2, similar trends hold even when only the mixer is removed.

All architectures reveal a striking pattern: while performance on knowledge-focused benchmarks declines steadily as layers are pruned, MMLU performance remains stable—until a single layer is removed, at which point it drops sharply. This suggests that *a specific layer encodes the critical skills required for MMLU*. High MMLU scores can persist even when most knowledge capacity is lost, highlighting the role of targeted retrieval components over general knowledge.

### 3.3. MMLU Relies on Two Crucial Layers

Based on insights from Appendix A.1, we refer to the *minimal model* as the smallest subnetwork that retains all layers up to (and including) the layer before the sharp drop in MMLU performance. We now repeat the previous experiment, this time *on the minimal model*, with a slight modification. Instead of cumulatively removing one layer at a time from the end of the network, we remove a single layer, evaluate the model performance without it, and then restore the layer before proceeding to the next.

Appendix A.3 illustrates the results of this experiment in Llama-3.1-8B, Falcon-Mamba-7B and Llamba-8B as well. The findings reveal a surprising insight on the minimal model: ***not one, but two layers are critical for the performance of MMLU***. Unlike earlier layers, the last two layers of the minimal model show no correlation between MMLU score and knowledge task performance. Pruning either layer causes a significant drop in MMLU accuracy while causing only a minor drop in scores on knowledge-focused tasks, underscoring their essential role in the task.

### 3.4. Zooming In: MMLU Relies on Two Crucial Heads

Building on Section 3.3, we now zoom in from layers to individual heads, analyzing how specific attention heads within the two critical layers affect model performance.

Using the minimal model setup from that section (with only the last two layers active), we ablated individual heads by zeroing their outputs and measured the effect on MMLU and knowledge-focused tasks. This revealed a small subset of heads whose removal sharply reduced MMLU accuracy while leaving other knowledge-focused benchmarks unaffected. We label these as *critical heads*.

To confirm their importance, we evaluated four configurations: (1) no critical heads retained, (2) only last-layer critical heads, (3) only second-last-layer critical heads, and (4) critical heads in both layers. The results presented in

Table 1 demonstrate that **MMLU depends on just a few mixing heads** rather than the model's broader knowledge representation. Disabling them drops performance to near-random. Remarkably, in Llama-3.1-8B, a single head is sufficient to achieve over 66% MMLU accuracy, and in Falcon-Mamba-8B, four Mamba-1 channels reach 52.26%.

### 3.5. Retrieval Emerges as a Key Factor in MMLU

Earlier results showed that two attention heads from separate layers work together to solve MMLU across architectures. We now provide evidence that these heads form part of a broader *in-context retrieval mechanism*, validated through a synthetic key-value (KV) retrieval task.

In this task, the model is given a dictionary and must recall the value of a queried key, for example,

```
Memorize the following dictionary:
present:50
institute:0
scallops:84
neuropsychiatry:67
The value of the key 'scallops' is
```

To succeed, the model must accurately store mappings (e.g., "scallops → 84") and, when queried, retrieve the correct value ('84') by identifying 'scallops' in its stored representations without confusion.

We tested 55 configurations with growing numbers of key-value pairs, each using 1,000 synthetic samples. For each configuration, we evaluated Llama-3.1-8B in three settings: intact, with L17H24 removed, and with L16H22 removed (the heads identified as critical in Section 3.4). The full evaluation setup and prompt format are discussed in Appendix A.9.

As shown in Figure 5, removing either L16H22 or L17H24 significantly impairs retrieval, confirming their central role. Notably, the impact of removing L17H24 increases with dictionary size, suggesting its contribution grows with task complexity. These findings extend previous research: the retrieval heads we identified align with and generalize the Content Gatherer head described by Lieberum et al. (2023) and the retrieval mechanisms documented by Wu et al. (2024). Importantly, our work demonstrates that similar retrieval mechanisms also emerge within SSM-based language models. The strong link between retrieval and MMLU performance suggests that the Transformer–SSM gap may stem from differences in a small number of retrieval-critical heads rather than fundamental architectural limitations. This reframes the performance gap as a targeted functional difference—where SSMs may lack specific retrieval capabilities—rather than a broad modeling inadequacy.

## 4. Gather-and-Aggregate Heads

We now examine how the critical retrieval heads identified in Section 3 across architectures implement the same Gather-and-Aggregate (G&A) mechanism. The mechanism unfolds in two stages. First, a **Gather Head** isolates meaningful segments within the input—such as multiple-choice answers—and summarizes each into a representative token. Then, a later **Aggregate Head** uses these summaries to retrieve the information most relevant to the prompt.

While previous work has described retrieval-oriented heads in Transformers—such as the "Correct Letter" heads identified by Lieberum et al. (2023)—these analyses focused narrowly on multiple-choice tasks. Our findings show that such heads are part of a broader retrieval strategy that generalizes beyond this format. This also refines the interpretation from Wu et al. (2024), who grouped retrieval heads under a single role; in contrast, we find they specialize in complementary functions within a coordinated process. Finally, we show that this same mechanism emerges in SSM-based models, despite their lack of explicit attention. These models exhibit smoother and less discrete patterns, but still learn to implement a G&A strategy using different architectural tools.

This section unpacks the mechanism in detail. We begin with how it operates in Transformers (Sections 4.1 and 4.2), then present experimental evidence that it also underlies retrieval in SSMs (Section 4.3)

### 4.1. Gather Heads in Transformers

A **Gather Head** compresses semantically related tokens within a segment into a single representative token. In Transformer models, this typically occurs through attention patterns that cause the final token of each segment to "attend to" all prior tokens in the same segment, effectively summarizing them into a compact representation (see Figure 1(a)).

For example, consider the multiple-choice format introduced in Section 3:

$$\boxed{\texttt{A. WPA\textbackslash n}} \rightarrow \texttt{A. WPA}\boxed{\texttt{\textbackslash n}}$$

Here, the newline token (\n) at the end of each answer choice acts as a summary token. This transformation reduces a segment from multiple tokens to one, simplifying downstream computation.

Lieberum et al. (2023) observed this behavior in 70B Chinchilla, where attention heads collect tokens like "A", ".", "W", "PA", and "\n" into the final newline token. The head's attention pattern essentially performs a form of local aggregation, summarizing the contents of each segment.

While effective, this process is not always clean: attention sometimes leaks to unrelated tokens—particularly "attention sinks" (Xiao et al., 2024) used to stabilize processing over long contexts.

### 4.2. Aggregate Heads in Transformers

Once the Gather Head condenses segments into representative tokens, an **Aggregate Head**—in a subsequent layer—retrieves and combines these summaries to inform the model's prediction. This head typically attends from a target token (e.g., the final "Answer:" position) over all gathered summaries, assigning greater weights to the most relevant one (see Figure 1(b)). This yields a sparse linear combination over segment summaries, often resembling an `argmax`-like operation (Tulchinskii et al., 2024):

$$\texttt{"Answer:"} = \boxed{\texttt{\textbackslash n}} + \boxed{\texttt{\textbackslash n}} + \boxed{\texttt{\textbackslash n}} + \boxed{\texttt{\textbackslash n}}$$

In Transformer models, such heads were previously termed "Correct Letter" Heads (Lieberum et al., 2023), due to their consistent weighting of the correct answer's summary token. However, the aggregation pattern generalizes beyond multiple-choice QA, enabling information retrieval in broader contexts.

As with Gather Heads, this mechanism is imperfect. Aggregate Heads sometimes distribute attention to irrelevant tokens, including attention sinks. These limitations become more pronounced in SSMs, as we discuss next.

### 4.3. G&A in Transformers and SSMs: Experimental Evidence

The G&A pattern is well-documented in Transformers—but do SSM models like Mamba implement the same structure? We test this via a two-part experiment that isolates and evaluates specific heads: (1) **Pruning:** We prune each model to a minimal configuration (see Section 3.3), zeroing out all heads in the final two layers except those suspected to implement G&A (see Table 1); (2) **Masking:** At runtime, we constrain the preserved heads' attention patterns: *Gather Heads* may only attend within each segment and to its final token, and *Aggregate Heads* may only attend to those final summary tokens (See Figure 6).

In all models—Llama-3.1-8B (Transformer), Llamba-8B (Mamba-2), and Falcon-Mamba-7B (Mamba-1)—the pruned and masked configurations match the original MMLU accuracy. This confirms that the selected heads are sufficient to drive retrieval and that the G&A mechanism is not unique to Transformers: despite architectural differences, SSM-based models converge on the same strategy.

Nevertheless, the attention patterns differ in texture. As shown in Figure 7, attention maps in Mamba-based models are smoother and less sharply localized. This softness limits their ability to isolate segments cleanly: *Gather Heads* in

SSMs often leak attention outside segment boundaries, and *Aggregate Heads* struggle to sharply focus on the correct summary token. These imperfections stem from the continuous nature of SSM hidden states and are explored further in Section 5.

## 5. The G&A Bottleneck in SSMs

The previous section established that both Transformers and SSMs rely on a small number of heads to implement a two-stage Gather-and-Aggregate (G&A) mechanism. We now examine how this mechanism breaks down in SSMs, providing a mechanistic account of their retrieval limitations.

We begin by visually inspecting G&A behavior in SSMs (Section 5.1), showing that Aggregate Heads attend broadly rather than selectively. We then quantify redundancy in SSM aggregation (Section 5.2), finding that multiple heads are needed to match the performance of fewer, stronger Transformer heads. Next, we show that hybrid models offload aggregation to attention layers (Section 5.3); ablating just six heads causes a $4\times$ drop in MMLU. We then identify these heads as the bottleneck (Section 5.4): replacing a single SSM layer restores MMLU from 33% to 50%. Finally, we explore how targeted attention during training or distillation can restore G&A in SSMs (Section 5.5).

### 5.1. Visual Inspection of G&A Heads

Figures 6 and 7 show that Mamba Aggregate Heads correctly attend to segment-final tokens but also attend to neighboring tokens unnecessarily. This likely introduces noise and reduces aggregation effectiveness. This smooth behavior reflects hidden-state architectures, where information compression over time makes it harder to capture the global nature of Aggregate Heads.

### 5.2. Redundancy of Aggregate Heads in SSM Models

We find that SSM-based models require more Aggregate Heads than Transformer-based models to reach similar or even lower levels of performance. This suggests that SSMs compensate for limited per-head expressiveness by distributing the retrieval task across multiple weaker components. Three observations support this claim:

1. **Gradual Performance Decay.** As shown in Table 4 and discussed in Section 6, ablating heads in Llamba causes a slow, steady drop in performance—unlike the sharp drops seen in Transformers—implying overlapping functionality and less specialization.

2. **Weaker, Less Isolated Heads.** Llamba-8B, distilled from Llama-3.1-8B via MOHAWK (Bick et al., 2024), adds G&A heads in layers 16 and 17. But while Llama-3.1-8B relies on two strong heads (L17H24 and L17H27),

Llamba-8B depends on a single moderate one (L17H31), with others contributing little. Rather than concentrating aggregation in a few strong heads, it spreads the load across weaker ones.

3. **More Heads Needed to Match Accuracy.** Llama-3.1-Minimal reaches 98% of its MMLU score with one Aggregate Head, Llamba-8B-Minimal needs two for 95%, and Falcon-Mamba-7B-Minimal requires four for 86% (see Table 1). This points to lower per-head capacity in SSMs.

### 5.3. Hybrid Models Delegate Aggregation to Attention

Attention layers in hybrid models are known to be critical for retrieval and copying tasks (Lieber et al., 2024; Glorioso et al., 2024; Waleffe et al., 2024). We hypothesize that during training, *Aggregate Heads naturally concentrate in the attention layers*, as attention's greater expressivity makes it better suited for aggregation.

To test this, we evaluate Zamba2-7B (Glorioso et al., 2024), the strongest 7B-scale hybrid model to date. Table 2 identifies attention heads critical for general retrieval; here, we further select nine that are especially important for MMLU: L47H{2, 4, 6, 8, 18, 21, 25} and L59H{17, 21}. Each head is disabled by zeroing out the final row of its attention matrix—a targeted intervention that impairs aggregation while leaving other capabilities, such as gathering and contextual mixing, largely intact. All SSM layers remain unchanged, preserving any aggregation they may contribute.

This intervention reduces MMLU accuracy from 64.3% to 34.9%, while knowledge task accuracy remains stable at 70%. Since MMLU's random baseline is 25%, this *4x* drop demonstrates that SSMs alone cannot sustain strong aggregation—hybrid models depend on attention layers to implement the G&A mechanism.

### 5.4. Hybrid Replacements

To further test whether SSMs struggle to implement Aggregate Heads, we examine how well an SSM mixer can replicate an attention layer. Using Llamba-8B (Bick et al., 2025), we distill each Mamba-2 layer independently from Llama-3.1-8B following Bick et al. (2024), aligning outputs via L2 loss without end-to-end optimization. Specifically, $i$-th Llamba block receives the output of Llama's $(i-1)$-th layer and matches Llama's $i$-th layer.

After assembling the full model from the individually distilled layers, the distilled model recovers 64% of Llama's 69% on knowledge tasks, while its MMLU performance remains 33%, compared to 67% for Llama. This suggests that **a key retrieval-related capability was not captured**.

To determine whether the observed performance gap is re-

lated to retrieval and specifically to the G&A mechanism, we conducted a *hybrid replacement experiment*. In this experiment, we systematically replaced each layer in the distilled Llamba-8B model with its corresponding layer from Llama-3.1-8B, evaluated its impact on MMLU (without fine-tuning), and then reverted the model before proceeding to the next layer.

The results, shown in Figure 8, reveal that most layer replacements had little or even a negative impact on MMLU. However, replacing L17, which contains two prominent G&A heads at H24 and H27 (L17H24 was identified in Table 1), significantly improved performance from 33% to 50%. To isolate their effect, we ran additional hybrid experiments. Swapping only L17H24, while keeping the rest of the SSM heads intact, raised performance to 44%. Including L17H27 as well increased it to 50%. This confirms that the contribution stems from the Aggregate Heads within L17, and further supports that individual SSM layers struggle to implement the Aggregate Head within G&A—reinforcing our claim that this limitation contributes to the performance gap between Transformers and SSMs.

### 5.5. Addressing G&A Limitations in SSM-Based Models

Efforts to improve G&A in pure SSMs face a core constraint: retrieval suffers due to fixed-size memory in RNN-style designs (Jelassi et al., 2024; Wen et al., 2024). Rather than working within these limits, we explore hybrids that introduce attention layers to target retrieval bottlenecks directly. Our analysis reveals two empirical patterns that guide hybrid model design:

1. **Training hybrids from scratch.** G&A heads consistently emerge in the middle of the network—for example, at layers L16 and L17 out of 32 in Llama-3.1-8B and Llamba-8B, and L35 and L36 out of 64 in Falcon-Mamba-8B. This suggests that attention is most valuable in mid layers, where retrieval operations naturally occur. As a result, when designing hybrid models from scratch, placing attention layers near the middle offers a targeted way to support G&A without relying on full attention throughout the network.

2. **Distilling hybrids from a Transformer.** As shown in Section 5.4, replacing individual layers in a distilled SSM with their attention-based counterparts from the teacher yields little benefit—except at L17, where MMLU improves from 33% to 50%. This indicates that the performance gap is concentrated in the Aggregate Head. Based on this, a practical strategy is to retain attention in the layers where strong G&A heads appear and distill the rest into SSMs. This allows the hybrid to recover retrieval quality where it matters most, while still benefiting from the efficiency of SSMs elsewhere.

These observations provide a practical recipe for hybrid design: introduce attention selectively—either guided by where G&A heads naturally emerge when training from scratch, or retained only in key locations when distilling from a Transformer. We leave further investigation into optimal placement and architectural variants of hybrid models to future work.

## 6. Transformer-SSM Gap Beyond MMLU

We previously showed that both Transformers and SSMs rely on a few heads to implement a two-stage Gather-and-Aggregate (G&A) mechanism for in-context retrieval. We now extend this analysis to more benchmarks and models, reinforcing that the broader retrieval gap stems from specific algorithmic limits rather than general language modeling ability. Our analysis has three parts: Section 6.1 identifies G&A heads across models; Section 6.2 evaluates how architectural choices affect performance; Section 6.3 examines how task format influences these effects.

### 6.1. Isolating Retrieval Differences Across Architectures

In Section 3, we showed that MMLU performance can be severely degraded by removing a single critical head—even when the model performs poorly on knowledge-focused tasks. This scenario is specific to MMLU: a task where strong retrieval alone can yield high accuracy, even when the model's knowledge is degraded. In contrast, most benchmarks require both retrieval and factual knowledge, making it harder to isolate the contribution of retrieval alone.

As a result, the minimal-model strategy used in MMLU fails on other tasks: pruning layers diminishes knowledge capacity and lowers scores across the board, masking the effect of retrieval-specific mechanisms. To avoid this confound, we instead take a different approach: rather than pruning the model, we preserve its knowledge and selectively disable only the G&A heads.

As detailed in Section 3.5, we use the synthetic KV-Retrieval task to identify G&A heads by ablating each one and measuring its effect on retrieval accuracy (Table 2). A performance drop signals the head's involvement in G&A. This method, also used by Wu et al. (2024), isolates retrieval ability without relying on external knowledge, enabling us to trace G&A heads across layers and architectures in a controlled, generalizable way.

### 6.2. Quantifying the Retrieval Gap

Having identified the retrieval-critical heads, we now measure their practical impact across benchmarks by incrementally disabling them in three model families. These include Transformer models (Llama-3.1-8B-Instruct and Llama-3.2-3B-Instruct (Touvron et al., 2023)) as baselines for effective

G&A implementation, pure SSM models (Llamba-3B and Llamba-8B (Bick et al., 2025)) based on Mamba-2, and hybrid models (Zamba2-2.7B and Zamba2-7B (Glorioso et al., 2024)) that use Mamba-2 layers with shared attention modules. For Zamba2, we ablate only attention heads, as they perform retrieval (Section 5.3). Each shared head is ablated in one instance to isolate its effect.

Table 4 summarizes the results. Transformer models show a sharp performance drop on retrieval-heavy tasks after removing just 10–20 heads, while knowledge-task accuracy remains stable—confirming retrieval is concentrated in a few heads. Zamba2-7B behaves similarly, with consistent effects across shared-head instances, reinforcing their unified G&A roles (Table 2). In contrast, Llamba-8B degrades slowly. Even after removing 30 G&A heads, it retains over 95% of its original performance. This gradual drop reflects redundancy: SSMs distribute retrieval across many weaker heads, unlike the focused G&A in Transformers—consistent with Section 5.

### 6.3. Amplifying the Retrieval Gap through Task Format

Beyond architecture, task format can amplify or reduce the retrieval gap between Transformers and SSMs. We previously noted this in MMLU, where models select a letter label rather than score full answer texts.

To further illustrate, we compare two formats of the same content: the original ARC-Challenge multiple-choice task and its conversational variant, ARC-Challenge Chat. As shown in Table 3, disabling G&A heads has little effect on the multiple-choice version but sharply reduces accuracy in the chat variant. The conversational format increases retrieval demands by spreading relevant content across dialogue turns, favoring architectures with sharper retrieval (Transformers and hybrids) over those with more diffuse mechanisms (pure SSMs).

This highlights a key point: retrieval pressure is shaped not just by what is asked, but by how it's presented. When context is more distributed, effective in-context retrieval becomes more critical—even if the underlying knowledge stays the same.

## 7. Conclusions

We show that both Transformer and SSM models develop a Gather-and-Aggregate (G&A) mechanism across a small number of heads to support in-context retrieval, which is crucial for performance on many benchmarks. While SSMs struggle to implement strong G&A heads due to their fixed-size hidden state, this limitation is localized and does not reflect a general weakness in language modeling.

Hybrid models provide further insight: they naturally of-fload G&A to attention layers, leveraging their stronger algorithmic capacity. This explains why even a few attention layers can meaningfully boost SSM models, providing practical guidance for hybrid design.

Our results also reframe benchmark interpretation. Although MMLU is often seen as a broad knowledge test, we find that MMLU performance is primarily governed by retrieval ability. This highlights the role of algorithmic skills, and particularly retrieval, in benchmark success.

Important open questions remain: while narrowing the retrieval gap is a key step, the role of G&A in in-context learning and copying warrants further study. It also remains to be seen whether G&A is part of a broader retrieval process involving other head types, such as Amplification Heads (Lieberum et al., 2023), which enhance signal propagation.

## Impact Statement

This paper presents work whose goal is to advance the field of Machine Learning. There are many potential societal consequences of our work, none which we feel must be specifically highlighted here.

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

# A. Complementary Material for Section 3

In this section, we present additional results from the experiments conducted on the models explored in Section 3. These results provide further insights into the behavior of knowledge and retrieval capabilities across layers and heads in different models.

## A.1. Detailed Performance Across Layers

We extend the analysis presented in Section 3.2 by providing performance data on other models, which show similar trends. As demonstrated in Appendix A.1, the pattern of stable MMLU scores despite a gradual decline in knowledge task performance is consistent across all models tested. Notably, the sharp drop in MMLU performance upon removal of specific layers reaffirms the criticality of certain layers in encoding retrieval skills.

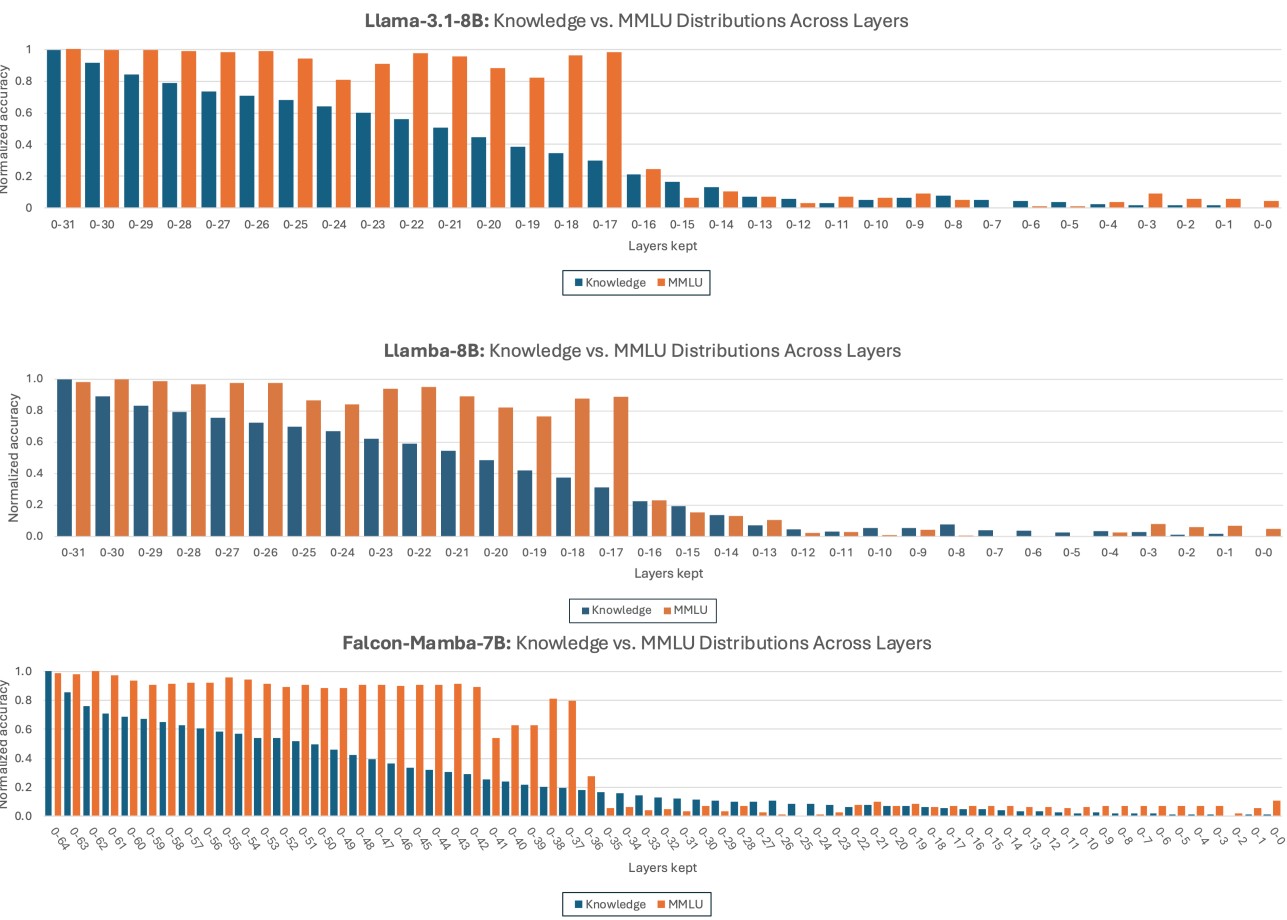

*Figure 2.* Residual performance trends for knowledge tasks and the MMLU benchmark as layers are progressively pruned layers of Llama-3.1-8B, Llamba-8B, and Falcon-Mamba-7B. The leftmost bar represents the full model with all layers intact, while subsequent bars correspond to models with layers pruned incrementally from the end of the network. The plot highlights three key observations: (1) MMLU scores remain relatively stable despite a decline in general knowledge tasks performance, indicating reliance on task-specific skills; (2) a sharp drop in MMLU performance identifies a critical layer encoding essential skills; and (3) the gradual decline in knowledge tasks' scores reflects the distributed nature of knowledge representation across layers.

## A.2. Layer Removal: Mixer and MLP Components

For both Llama-3.1-8B and Llamba-8B, we performed experiments where entire layers, including both the mixer and MLP components, were removed. This approach was taken under the assumption that MLP components likely depend on outputs

from the mixer components, which are crucial for information flow across layers. The results from these tests, presented in Appendix A.2, show that similar performance trends are observed whether the MLP components are retained or not. This suggests that the mixer components play a more prominent role in encoding retrieval capabilities, while the MLP components do not significantly alter the overall performance on knowledge tasks and MMLU benchmarks. This finding reinforces the idea that retrieval skills are primarily concentrated in specific layers and do not rely heavily on the MLP components within those layers.

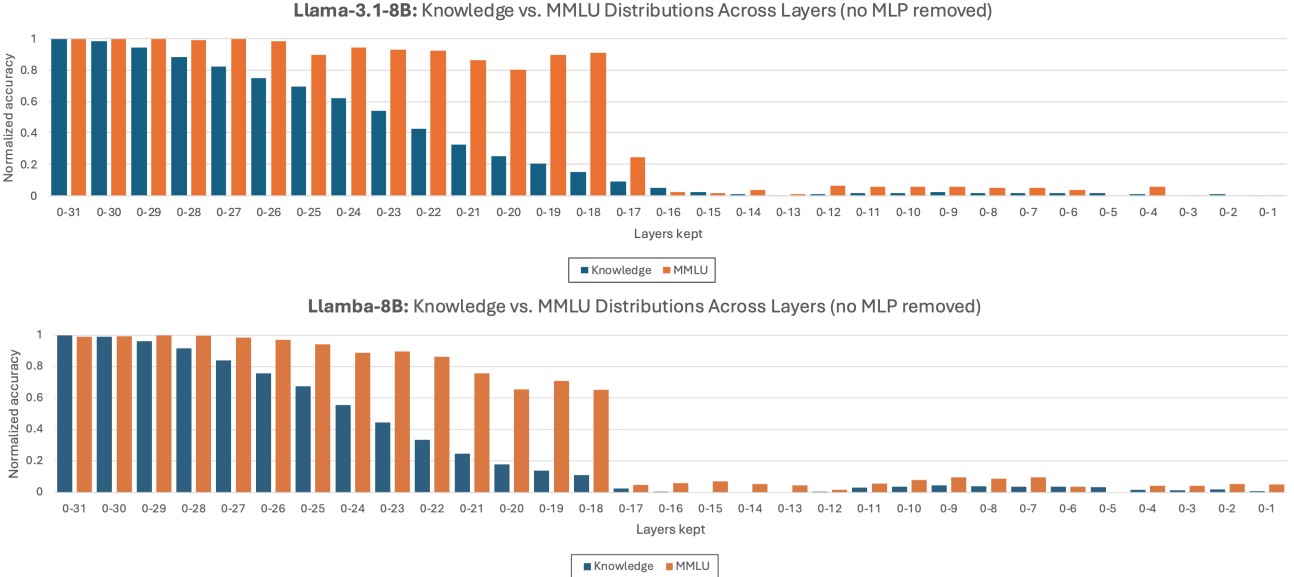

*Figure 3.* Residual performance trends for knowledge tasks and the MMLU benchmark as layers are progressively pruned from Llama-3.1-8B, Llamba-8B but keeping the MLPs intact. This plot yields similar results to Appendix A.1

### A.3. Performance in Minimal Models

The results from Section 3.3 are further elaborated in Appendix A.3. The performance of Llama-3.1-8B and other models when subjected to layer removal shows that, for all models, not one, but two layers are crucial for MMLU performance. The surprising robustness of MMLU performance in the minimal models highlights the importance of fine-tuning layer-wise architecture to achieve strong retrieval performance while retaining knowledge capabilities.

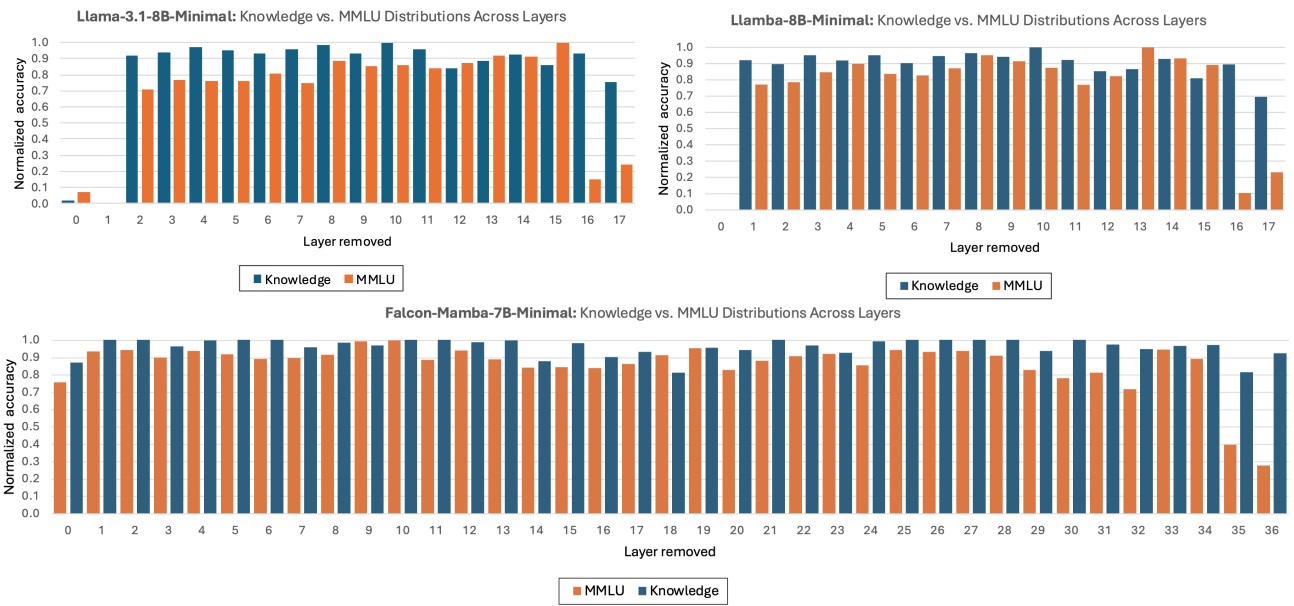

*Figure 4.* Residual performance trends for the MMLU benchmark and knowledge tasks metrics in the minimal model (defined with respect to MMLU) for Llama-3.1-8B, Llamba-8B, and Falcon-Mamba-7B. In this experiment, each layer is independently pruned, and the resulting performance is assessed. The results reveal that *two layers are critical for MMLU performance*, as pruning either of the last two layers causes a significant drop in MMLU scores while leaving knowledge metrics largely unaffected. This finding highlights the unique role of these layers in task-specific skills rather than general knowledge representation.

## A.4. Critical Heads Drive MMLU Performance Across Models

To assess how in-context retrieval is localized within specific heads, we examine the MMLU performance of three models—Llama-3.1-8B, Llamba-8B, and Falcon-Mamba-7B—under targeted head ablation. Table 1 presents the results of selectively retaining or removing mixing heads in the final two layers of each model, while preserving all earlier components.

Across all architectures, we observe a sharp and consistent pattern: removing just one or two critical heads reduces MMLU accuracy to near-random levels (∼25%), while retaining them restores much of the original performance. Notably, these heads represent fewer than 0.1% of the total head count, highlighting the extreme concentration of retrieval ability within a tiny subset of the model. Meanwhile, scores on knowledge-focused benchmarks remain nearly unchanged across all configurations, reinforcing that these heads are specifically responsible for in-context retrieval rather than general language modeling or factual recall.

This result supports the broader claim that the Gather-and-Aggregate mechanism emerges in just a handful of components and suggests that hybrid and efficient model designs can selectively preserve only these heads.

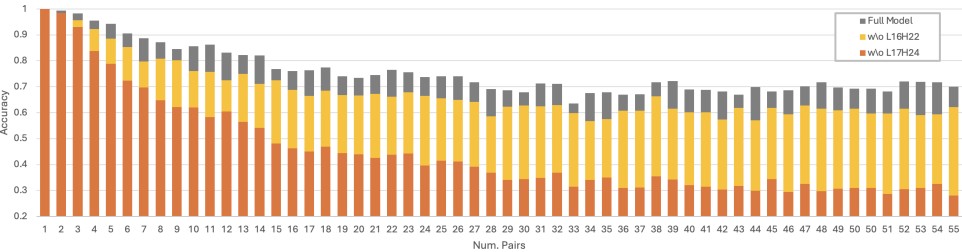

*Figure 5.* **Evaluation of Llama-3.1-8B on the KV-Retrieval task under three settings: unmodified, with only `L16H22` removed, or with only `L17H24` removed (the Gather and Aggregate Heads identified in Section 3.4, respectively).** Performance is reported across 55 configurations with increasing numbers of key-value pairs, using the original (non-pruned) model. Each bar shows accuracy for a given dictionary size—for example, with 20 pairs, the original score is 85.6%, removing only the Gather Head (`L16H22`) drops accuracy to 76%, and removing only the Aggregate Head (`L17H24`) drops it to 62%. These results underscore two key points we build on later: (1) Retrieval is more sensitive to Aggregate Head removal than to Gather Head removal, underscoring its importance; (2) The Aggregate Head becomes increasingly important as task complexity grows.

| Model | Heads Retained | | | Metrics (%) | |
|---|---|---|---|---|---|
| | **Prev. Layers** | **Layer X** | **Layer X+1** | **MMLU** | **Knowledge Tasks** |
| Llama-3.1-8B | 0–31 | 22 | 24 | **66.32** | 39.09 |
| | 0–31 | ∅ | 24 | 24.36 | 39.18 |
| | 0–31 | 22 | ∅ | 25.59 | 39.21 |
| | 0–31 | ∅ | ∅ | 25.56 | 39.21 |
| Llamba-8B | 0–31 | 10 | 3, 9 | **58.92** | 39.08 |
| | 0–31 | ∅ | 3, 9 | 42.17 | 39.16 |
| | 0–31 | 10 | ∅ | 38.02 | 39.00 |
| | 0–31 | ∅ | ∅ | 29.82 | 36.76 |
| Falcon-Mamba-7B | 0–8191 | 3186, 5143 6607, 7305 | 1565, 1906 3873, 4925 | **52.26** | 37.23 |
| | 0–8191 | ∅ | 1565, 1906 3873, 4925 | 35.91 | 37.25 |
| | 0–8191 | 3186, 5143 6607, 7305 | ∅ | 39.39 | 37.26 |
| | 0–8191 | ∅ | ∅ | 24.98 | 37.28 |

*Table 1.* **Impact of Critical Head Removal Across Model Architectures.** We evaluate how retaining or removing specific heads in the final two layers affects MMLU and knowledge task performance for three models: Llama-3.1-8B, Llamba-8B, and Falcon-Mamba-7B. Each row shows performance under a different configuration, where the "Heads Retained" columns indicate which heads are active in the two final layers ("Layer X" and "Layer X+1") while all earlier heads are preserved. The heads shown correspond to the critical Gather and Aggregate Heads identified in prior analyses. For Llama and Llamba models, retaining both heads restores near-original MMLU performance (66.3% and 58.9%, respectively), while removing either or both causes performance to collapse to near-random levels. In Falcon-Mamba-7B, just four attention channels (among thousands) across the final two layers are sufficient to recover 52.3% MMLU accuracy—again, comparable to full-model performance. Knowledge task scores remain mostly unchanged across configurations, reinforcing that these heads specifically control retrieval rather than factual knowledge access. This shows that across architectures, in-context retrieval is driven by just a few heads, whose removal alone accounts for the majority of the performance gap.

## A.5. Retrieval Heads Across Model Families

Table 2 presents the 50 most retrieval-relevant heads for each model, ranked by their individual contribution to KV-Retrieval accuracy. Several trends emerge. In Transformers, top heads cluster in upper layers, often overlapping with those critical for MMLU. In SSM-based models, retrieval heads are more diffuse and individually weaker. Zamba2 hybrids concentrate high-performing heads around shared-attention layers, suggesting these layers anchor retrieval in otherwise SSM-heavy architectures.

*Table 2.* **Attention Head Ablation Study:** Performance impact on KV-Retrieval task when individual attention heads are removed from various language models. Each head was systematically knocked out and the model's accuracy was evaluated on a KV-Retrieval benchmark with dictionary size of 20 across 1000 samples. After measurement, each head was restored before testing the next one. Results show the **50 most significant attention heads** from each model, sorted by accuracy, revealing which heads contribute most critically to the KV-Retrieval capability across Zamba2, Llama, and Llamba model families. For Zamba2 models, $^{\dagger}$ marks layers with the first shared attention, and $^{\ddagger}$ marks those with the second.

| Zamba2-2.7B | | | Zamba2-7B | | | Llama-3.2-3B | | | Llama-3.1-8B | | | Llamba-3B | | | Llamba-8B | | |
|---|---|---|---|---|---|---|---|---|---|---|---|---|---|---|---|---|---|
| Layer | Head | Acc | Layer | Head | Acc | Layer | Head | Acc | Layer | Head | Acc | Layer | Head | Acc | Layer | Head | Acc |
| $6^{\dagger}$ | 0 | 0.088 | $11^{\ddagger}$ | 28 | 0.139 | 14 | 7 | 0.381 | 12 | 1 | 0.421 | 20 | 30 | 0.214 | 17 | 31 | 0.382 |
| $18^{\dagger}$ | 12 | 0.199 | $23^{\ddagger}$ | 28 | 0.139 | 12 | 1 | 0.405 | 8 | 8 | 0.494 | 1 | 0 | 0.220 | 21 | 2 | 0.446 |
| $6^{\dagger}$ | 12 | 0.303 | $35^{\ddagger}$ | 28 | 0.139 | 14 | 6 | 0.524 | 13 | 5 | 0.514 | 15 | 30 | 0.296 | 16 | 29 | 0.460 |
| $30^{\dagger}$ | 0 | 0.328 | $47^{\ddagger}$ | 28 | 0.139 | 15 | 18 | 0.545 | 14 | 7 | 0.519 | 14 | 17 | 0.301 | 22 | 26 | 0.465 |
| $6^{\dagger}$ | 26 | 0.380 | $59^{\ddagger}$ | 28 | 0.176 | 13 | 5 | 0.589 | 13 | 17 | 0.596 | 11 | 23 | 0.317 | 27 | 15 | 0.471 |
| $30^{\dagger}$ | 4 | 0.421 | $71^{\ddagger}$ | 28 | 0.176 | 10 | 17 | 0.597 | 15 | 18 | 0.635 | 12 | 24 | 0.320 | 14 | 9 | 0.471 |
| $18^{\dagger}$ | 25 | 0.426 | $11^{\ddagger}$ | 21 | 0.266 | 9 | 1 | 0.625 | 14 | 6 | 0.680 | 19 | 13 | 0.324 | 11 | 25 | 0.476 |
| $30^{\dagger}$ | 20 | 0.430 | $23^{\ddagger}$ | 21 | 0.266 | 0 | 4 | 0.639 | 13 | 23 | 0.685 | 19 | 7 | 0.327 | 21 | 27 | 0.482 |
| $18^{\dagger}$ | 26 | 0.433 | $35^{\ddagger}$ | 21 | 0.266 | 12 | 13 | 0.648 | 12 | 2 | 0.730 | 10 | 20 | 0.329 | 16 | 18 | 0.487 |
| $30^{\dagger}$ | 31 | 0.472 | $47^{\ddagger}$ | 21 | 0.266 | 9 | 18 | 0.655 | 5 | 14 | 0.732 | 15 | 6 | 0.333 | 11 | 31 | 0.487 |
| $24^{\dagger}$ | 23 | 0.479 | $59^{\ddagger}$ | 21 | 0.281 | 12 | 8 | 0.657 | 10 | 17 | 0.761 | 24 | 17 | 0.334 | 10 | 15 | 0.490 |
| $30^{\dagger}$ | 12 | 0.482 | $71^{\ddagger}$ | 21 | 0.281 | 14 | 0 | 0.667 | 11 | 9 | 0.784 | 13 | 29 | 0.335 | 14 | 5 | 0.492 |
| $30^{\dagger}$ | 24 | 0.483 | $11^{\ddagger}$ | 3 | 0.424 | 15 | 20 | 0.670 | 5 | 22 | 0.816 | 7 | 28 | 0.337 | 11 | 29 | 0.492 |
| $30^{\dagger}$ | 26 | 0.488 | $23^{\ddagger}$ | 3 | 0.424 | 10 | 3 | 0.671 | 14 | 0 | 0.828 | 10 | 16 | 0.338 | 15 | 16 | 0.495 |
| $36^{\ddagger}$ | 3 | 0.496 | $35^{\ddagger}$ | 3 | 0.424 | 3 | 18 | 0.677 | 19 | 7 | 0.830 | 14 | 6 | 0.340 | 12 | 9 | 0.495 |
| $36^{\ddagger}$ | 30 | 0.496 | $47^{\ddagger}$ | 3 | 0.424 | 19 | 7 | 0.677 | 9 | 23 | 0.839 | 12 | 25 | 0.344 | 18 | 30 | 0.497 |
| $30^{\dagger}$ | 16 | 0.499 | $6^{\dagger}$ | 2 | 0.430 | 8 | 8 | 0.685 | 7 | 16 | 0.844 | 11 | 16 | 0.348 | 14 | 7 | 0.497 |
| $18^{\dagger}$ | 16 | 0.500 | $17^{\dagger}$ | 2 | 0.430 | 11 | 10 | 0.689 | 12 | 23 | 0.846 | 14 | 9 | 0.350 | 28 | 28 | 0.498 |
| $6^{\dagger}$ | 16 | 0.508 | $29^{\dagger}$ | 2 | 0.430 | 12 | 0 | 0.689 | 10 | 11 | 0.855 | 19 | 0 | 0.352 | 22 | 28 | 0.498 |
| $24^{\ddagger}$ | 3 | 0.514 | $41^{\dagger}$ | 2 | 0.430 | 13 | 16 | 0.693 | 16 | 15 | 0.855 | 13 | 19 | 0.352 | 9 | 28 | 0.501 |
| $24^{\ddagger}$ | 6 | 0.521 | $53^{\dagger}$ | 2 | 0.430 | 9 | 21 | 0.696 | 11 | 11 | 0.859 | 10 | 17 | 0.352 | 22 | 9 | 0.502 |
| $24^{\ddagger}$ | 7 | 0.521 | $65^{\dagger}$ | 2 | 0.434 | 11 | 1 | 0.698 | 7 | 4 | 0.860 | 16 | 22 | 0.353 | 9 | 20 | 0.502 |
| $30^{\dagger}$ | 25 | 0.524 | $77^{\dagger}$ | 2 | 0.434 | 12 | 12 | 0.699 | 12 | 7 | 0.868 | 12 | 10 | 0.353 | 11 | 0 | 0.504 |
| $24^{\ddagger}$ | 21 | 0.526 | $11^{\ddagger}$ | 15 | 0.473 | 3 | 2 | 0.703 | 10 | 20 | 0.870 | 2 | 9 | 0.355 | 8 | 17 | 0.504 |
| $18^{\dagger}$ | 20 | 0.532 | $11^{\ddagger}$ | 18 | 0.473 | 1 | 17 | 0.705 | 3 | 4 | 0.878 | 0 | 7 | 0.355 | 15 | 9 | 0.505 |
| $6^{\dagger}$ | 1 | 0.532 | $23^{\ddagger}$ | 15 | 0.473 | 5 | 13 | 0.705 | 0 | 7 | 0.879 | 1 | 23 | 0.356 | 6 | 10 | 0.505 |
| $24^{\ddagger}$ | 25 | 0.533 | $23^{\ddagger}$ | 18 | 0.473 | 13 | 22 | 0.705 | 4 | 5 | 0.883 | 8 | 26 | 0.357 | 5 | 24 | 0.505 |
| $12^{\dagger}$ | 12 | 0.534 | $35^{\ddagger}$ | 15 | 0.473 | 9 | 3 | 0.710 | 15 | 20 | 0.883 | 0 | 18 | 0.357 | 19 | 24 | 0.506 |
| $24^{\ddagger}$ | 22 | 0.538 | $35^{\ddagger}$ | 18 | 0.473 | 6 | 4 | 0.713 | 1 | 7 | 0.884 | 11 | 24 | 0.359 | 15 | 17 | 0.506 |
| $36^{\ddagger}$ | 16 | 0.539 | $47^{\ddagger}$ | 15 | 0.473 | 10 | 18 | 0.713 | 3 | 23 | 0.885 | 16 | 9 | 0.360 | 12 | 18 | 0.506 |
| $24^{\ddagger}$ | 13 | 0.539 | $47^{\ddagger}$ | 18 | 0.473 | 5 | 19 | 0.714 | 10 | 10 | 0.886 | 13 | 30 | 0.360 | 7 | 11 | 0.507 |
| $18^{\dagger}$ | 14 | 0.541 | $59^{\ddagger}$ | 3 | 0.480 | 12 | 10 | 0.717 | 12 | 3 | 0.886 | 11 | 0 | 0.352 | 24 | 5 | 0.508 |
| $30^{\dagger}$ | 6 | 0.544 | $71^{\ddagger}$ | 3 | 0.480 | 11 | 15 | 0.721 | 12 | 6 | 0.886 | 13 | 19 | 0.352 | 13 | 29 | 0.508 |
| $18^{\dagger}$ | 21 | 0.545 | $11^{\ddagger}$ | 17 | 0.488 | 9 | 17 | 0.722 | 9 | 11 | 0.888 | 7 | 7 | 0.360 | 7 | 20 | 0.508 |
| $12^{\ddagger}$ | 19 | 0.545 | $23^{\ddagger}$ | 17 | 0.488 | 4 | 4 | 0.726 | 11 | 1 | 0.890 | 6 | 1 | 0.360 | 7 | 30 | 0.508 |
| $18^{\dagger}$ | 19 | 0.546 | $35^{\ddagger}$ | 17 | 0.488 | 11 | 2 | 0.729 | 8 | 17 | 0.892 | 26 | 6 | 0.361 | 6 | 18 | 0.508 |
| $24^{\ddagger}$ | 15 | 0.547 | $47^{\ddagger}$ | 17 | 0.488 | 12 | 4 | 0.730 | 1 | 15 | 0.895 | 23 | 16 | 0.361 | 5 | 8 | 0.508 |
| $18^{\dagger}$ | 24 | 0.548 | $59^{\ddagger}$ | 15 | 0.488 | 10 | 8 | 0.732 | 4 | 4 | 0.896 | 19 | 9 | 0.361 | 13 | 31 | 0.509 |
| $6^{\dagger}$ | 25 | 0.548 | $71^{\ddagger}$ | 15 | 0.488 | 13 | 21 | 0.733 | 6 | 15 | 0.896 | 7 | 9 | 0.361 | 12 | 15 | 0.509 |
| $18^{\dagger}$ | 0 | 0.549 | $11^{\ddagger}$ | 10 | 0.496 | 2 | 16 | 0.735 | 9 | 18 | 0.896 | 22 | 27 | 0.362 | 11 | 12 | 0.509 |
| $6^{\dagger}$ | 11 | 0.549 | $23^{\ddagger}$ | 10 | 0.496 | 10 | 5 | 0.735 | 8 | 12 | 0.897 | 16 | 16 | 0.362 | 10 | 1 | 0.509 |
| $24^{\ddagger}$ | 12 | 0.550 | $35^{\ddagger}$ | 10 | 0.496 | 11 | 14 | 0.735 | 24 | 15 | 0.897 | 11 | 8 | 0.362 | 10 | 8 | 0.509 |
| $6^{\dagger}$ | 24 | 0.550 | $47^{\ddagger}$ | 10 | 0.496 | 11 | 9 | 0.739 | 3 | 3 | 0.898 | 9 | 16 | 0.362 | 10 | 17 | 0.509 |
| $18^{\dagger}$ | 2 | 0.551 | $59^{\ddagger}$ | 18 | 0.496 | 5 | 17 | 0.740 | 0 | 1 | 0.900 | 3 | 27 | 0.362 | 8 | 30 | 0.509 |
| $24^{\ddagger}$ | 1 | 0.552 | $71^{\ddagger}$ | 18 | 0.496 | 13 | 12 | 0.741 | 5 | 6 | 0.900 | 11 | 22 | 0.363 | 7 | 24 | 0.509 |
| $12^{\ddagger}$ | 9 | 0.552 | $6^{\dagger}$ | 1 | 0.500 | 9 | 8 | 0.743 | 20 | 11 | 0.900 | 6 | 24 | 0.363 | 5 | 7 | 0.509 |
| $6^{\dagger}$ | 22 | 0.553 | $17^{\dagger}$ | 1 | 0.500 | 7 | 3 | 0.744 | 3 | 16 | 0.901 | 18 | 7 | 0.364 | 12 | 7 | 0.510 |
| $18^{\dagger}$ | 6 | 0.554 | $29^{\dagger}$ | 1 | 0.500 | 12 | 22 | 0.745 | 10 | 4 | 0.902 | 16 | 15 | 0.364 | 11 | 8 | 0.510 |
| $12^{\ddagger}$ | 7 | 0.554 | $41^{\dagger}$ | 1 | 0.500 | 19 | 1 | 0.747 | 1 | 0 | 0.903 | 14 | 27 | 0.364 | 9 | 0 | 0.510 |
| $30^{\dagger}$ | 2 | 0.555 | $53^{\dagger}$ | 1 | 0.500 | 4 | 8 | 0.748 | 3 | 1 | 0.905 | 13 | 24 | 0.364 | 22 | 31 | 0.511 |

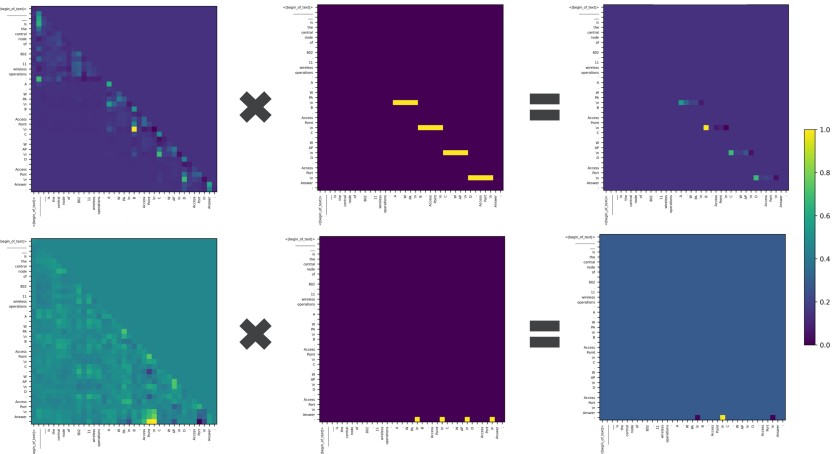

*Figure 6.* **Visualization of the masking applied to the Gather Head (first row) and the Aggregate Head (second row) of Llamba-8B**. The Gather Heads are restricted to interactions with the representative token and its associated answer, while the Aggregate Heads are limited to processing only the final token's relevant portions. This setup ensures that the hypothesized Gather-and-Aggregate mechanism is exclusively responsible for the observed model behavior during MMLU evaluation.

### A.6. Masked Head Evaluation Confirms Role Specialization

To confirm that the observed Gather-and-Aggregate behavior is not incidental, we conduct controlled masking experiments that restrict head input and output pathways. Figure 6 visualizes the masking applied to Llamba-8B: the Gather Head is constrained to interact only with a specific context segment (e.g., the representative token and its candidate answers), while the Aggregate Head is permitted to read only the final token. This enforces a clean separation of responsibilities and eliminates confounding interactions.

The results demonstrate that even under these constraints, performance remains high, indicating that the heads in question genuinely specialize in retrieval. Figure 7 further supports this by comparing attention maps across Transformer and SSM models. While Llama-3.1-8B shows sharp, focused patterns for both head types, SSM models exhibit smoother, more diffuse behavior—suggesting that explicit retrieval pathways are more difficult to encode in state-space models, but can still emerge under the right inductive bias or hybridization.

### A.7. Hybrid Layer Replacement Pinpoints Retrieval Bottlenecks

To further isolate the limitations of SSMs in implementing retrieval behavior, we perform layer-wise replacement experiments, shown in Figure 8. In this setup, we substitute individual Llamba-8B layers with their Llama-3.1-8B counterparts, then evaluate the resulting hybrid model on MMLU. Most substitutions have little or even negative effect, confirming that localized replacements do not trivially enhance retrieval. However, replacing Layer 17 yields a substantial boost—nearly doubling MMLU accuracy relative to the Llamba baseline.

This layer aligns with the Aggregate Head L17H24, previously identified as critical. The result reinforces our earlier claim: SSMs struggle particularly with the global integration stage of retrieval, and improving only this step—while leaving the rest of the model unchanged—can already unlock large performance gains.

### A.8. Disabling G&A Heads Impairs Reasoning But Not Knowledge Access

The head ablation results in Table 3 and Table 4 provide complementary evidence that Gather-and-Aggregate heads are responsible for task-specific reasoning rather than knowledge access alone. In ARC-Challenge, where answers are selected from fixed options, head removal causes modest degradation. But in ARC-Challenge-Chat, where generative reasoning is required, accuracy drops sharply with G&A removal.

Table 4 generalizes this across six benchmarks. MMLU, LAMBADA, GSM8K, and BBH show substantial performance drops—especially in models like Llama-8B and Zamba2—despite only moderate declines in knowledge-based tasks. These

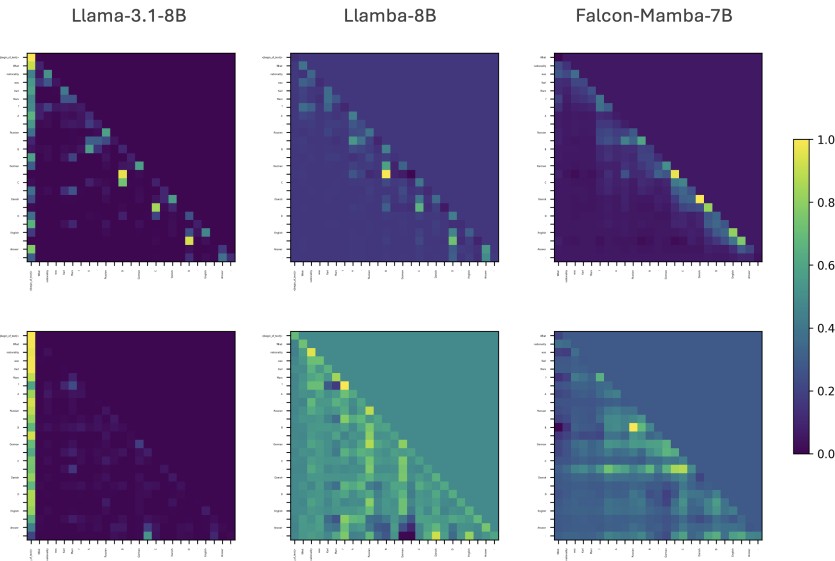

*Figure 7.* **Illustration of Gather Heads (top row) and Aggregate Heads (bottom row) across different model types.** The left column shows attention patterns in **Llama-3.1-8B (Transformer-based)**, while the center and right columns correspond to **Llamba-8B (Mamba-2-based)** and **Falcon-Mamba-7B (Mamba-1-based)**, respectively—both state-space models (SSMs). In Transformer models, both Gather and Aggregate Heads produce sharp, localized patterns. In contrast, SSM-based models exhibit smoother behavior, making it harder to isolate specific segments or retrieve information with the same precision.

results demonstrate that disabling G&A heads selectively damages retrieval and reasoning, without erasing stored knowledge. This aligns with our core hypothesis: these heads act as interfaces for *using* knowledge in context, not for *storing* it.

| MODEL | #REMOVED HEADS | ARC-CHALLENGE (CHAT) ACC ↑ | ARC-CHALLENGE (REGULAR) ACC ↑ |
|---|---|---|---|
| Llama-3B | 0 | 76.8 (+0.0%) | 45.5 (+0.0%) |
| | 10 | 72.2 (-6.0%) | 43.6 (-4.2%) |
| | 20 | 50.0 (-34.9%) | 42.0 (-7.7%) |
| | 30 | 43.2 (-43.8%) | 41.9 (-7.9%) |
| Llama-8B | 0 | 84.3 (+0.0%) | 54.9 (+0.0%) |
| | 10 | 77.1 (-8.5%) | 51.6 (-6.0%) |
| | 20 | 49.3 (-41.5%) | 47.3 (-13.8%) |
| | 30 | 53.6 (-36.4%) | 47.9 (-12.8%) |

*Table 3.* **Comparison of ARC Challenge and ARC-Challenge Chat under Attention Head Removal.** ARC-Challenge Chat repackages the same questions as ARC Challenge into a conversational, generative format. While ARC Challenge uses normalized accuracy, the chat version relies on exact-match due to its open-ended outputs. Models initially perform better on the chat variant, leveraging its generative flexibility to reason through questions. But when G&A heads are ablated, performance drops sharply—unlike the minor declines seen in ARC Challenge. This contrast suggests the degradation reflects impaired retrieval rather than loss of stored knowledge.

## A.9. KV-Retrieval Recipe

To probe retrieval behavior in isolation, we design a synthetic memorization task in which the model is asked to recall values associated with keys presented earlier in the same prompt. Each input contains a list of key-value (KV) pairs, followed by a query for the value of one of the keys. For example:

```
Memorize the following dictionary:
present:50
institute:0
```

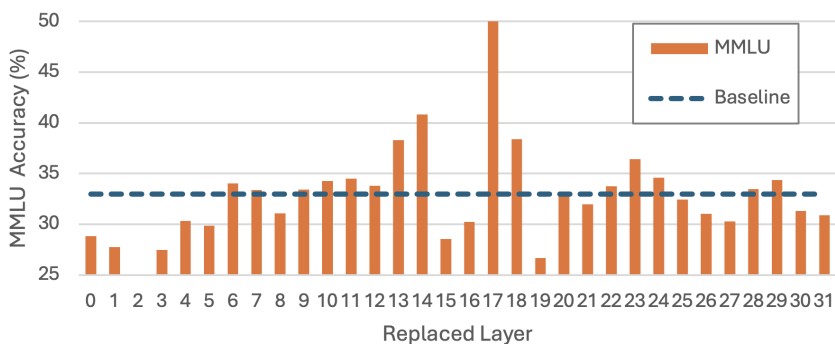

*Figure 8.* **Results of the hybrid replacement experiment.** Each Llamba-8B layer was replaced with its Llama-3.1-8B counterpart and tested on MMLU. Most swaps had little or negative impact on the 33% baseline, but replacing layer 17—where a key Aggregate Head is (Table 1)—significantly improved MMLU. This underscores both the difficulties SSMs face in replicating the Aggregate mechanism and G&A critical role in MMLU performance.

| MODEL | #HEADS | MMLU ACC ↑ | LAMB. PPL ↓ | GSM8K ACC ↑ | SWDE ACC ↑ | BBH ACC ↑ | KNOWLEDGE ACC ↑ |
|---|---|---|---|---|---|---|---|
| Llama-3B | 0 | 60.3 (+0.0%) | 4.8 (+0.0%) | 28.7 (+0.0%) | 85.8 (+0.0%) | 38.2 (+0.0%) | 60.5 (+0.0%) |
| | 10 | 53.1 (-12.0%) | 6.5 (+35.7%) | 17.4 (-39.4%) | 81.9 (-4.5%) | 33.4 (-12.6%) | 59.4 (-1.8%) |
| | 20 | 32.2 (-46.6%) | 8.8 (+82.8%) | 9.1 (-68.2%) | 57.5 (-33.0%) | 27.7 (-27.5%) | 58.7 (-3.0%) |
| | 30 | 29.9 (-50.4%) | 10.1 (+109%) | 5.6 (-80.5%) | 47.5 (-44.6%) | 25.4 (-33.5%) | 58.0 (-4.1%) |
| Llama-8B | 0 | 68.1 (+0.0%) | 3.4 (+0.0%) | 27.3 (+0.0%) | 90.8 (+0.0%) | 45.1 (+0.0%) | 68.5 (+0.0%) |
| | 10 | 61.9 (-9.1%) | 4.2 (+22.0%) | 21.7 (-20.5%) | 87.3 (-3.9%) | 37.7 (-16.5%) | 67.1 (-2.0%) |
| | 20 | 38.1 (-44.0%) | 6.8 (+98.6%) | 9.4 (-65.6%) | 79.5 (-12.4%) | 29.2 (-35.2%) | 64.8 (-5.4%) |
| | 30 | 38.7 (-43.2%) | 7.3 (+115%) | 7.8 (-71.4%) | 74.0 (-18.5%) | 29.0 (-35.7%) | 64.4 (-6.0%) |
| Llamba-3B | 0 | 52.5 (+0.0%) | 3.6 (+0.0%) | - | 21.3 (+0.0%) | 9.2 (+0.0%) | 63.8 (+0.0%) |
| | 10 | 42.6 (-18.9%) | 5.2 (+44.4%) | - | 18.6 (-12.7%) | 9.0 (-2.2%) | 63.7 (-0.2%) |
| | 20 | 41.3 (-21.3%) | 8.2 (+128%) | - | 18.1 (-15.0%) | 9.0 (-2.2%) | 63.1 (-1.1%) |
| | 30 | 41.2 (-21.5%) | 9.1 (+153%) | - | 18.1 (-15.0%) | 9.0 (-2.2%) | 62.6 (-1.9%) |
| Llamba-8B | 0 | 60.7 (+0.0%) | 4.09 (+0.0%) | - | 20.0 (+0.0%) | 11.0 (+0.0%) | 69.1 (+0.0%) |
| | 10 | 59.6 (-1.8%) | 6.26 (+53.2%) | - | 18.3 (-8.5%) | 11.6 (+5.5%) | 68.8 (-0.4%) |
| | 20 | 59.0 (-2.8%) | 6.99 (+70.8%) | - | 18.6 (-7.0%) | 11.4 (+3.6%) | 68.6 (-0.7%) |
| | 30 | 58.1 (-4.3%) | 7.44 (+81.9%) | - | 18.1 (-9.5%) | 11.4 (+3.6%) | 68.2 (-1.3%) |
| Zamba2-2.7B | 0 | 55.7 (+0.0%) | 4.2 (+0.0%) | 57.4 (+0.0%) | 89.5 (+0.0%) | 30.6 (+0.0%) | 66.8 (+0.0%) |
| | 10 | 42.4 (-23.9%) | 12.8 (+204%) | 24.7 (-57.0%) | 84.3 (-5.8%) | 25.5 (-16.7%) | 64.8 (-3.0%) |
| | 20 | 37.2 (-33.2%) | 22.2 (+428%) | 6.5 (-88.7%) | 74.4 (-16.9%) | 17.4 (-43.1%) | 62.6 (-6.3%) |
| Zamba2-7B | 0 | 65.1 (+0.0%) | 3.1 (+0.0%) | 60.5 (+0.0%) | 91.7 (+0.0%) | 33.0 (+0.0%) | 70.6 (+0.0%) |
| | 10 | 63.0 (-3.2%) | 3.6 (+16.1%) | 40.1 (-33.7%) | 80.0 (-12.8%) | 24.6 (-25.5%) | 68.7 (-2.7%) |
| | 20 | 57.3 (-12.0%) | 5.2 (+67.7%) | 27.6 (-54.4%) | 75.1 (-18.1%) | 28.9 (-12.4%) | 67.5 (-4.4%) |
| | 30 | 54.0 (-17.1%) | 7.0 (+126%) | 18.4 (-69.6%) | 43.7 (-52.3%) | 24.0 (-27.3%) | 67.3 (-4.7%) |
| | 40 | 50.6 (-22.3%) | 9.5 (+206%) | 14.9 (-75.4%) | 41.2 (-55.1%) | 21.7 (-34.2%) | 67.0 (-5.1%) |
| | 50 | 37.7 (-42.1%) | 13.9 (+348%) | 7.5 (-87.6%) | 39.6 (-56.8%) | 18.5 (-43.9%) | 67.0 (-5.1%) |
| | 60 | 36.2 (-44.4%) | 19.8 (+538%) | 7.2 (-88.1%) | 39.6 (-56.8%) | 15.9 (-51.8%) | 66.5 (-5.8%) |

*Table 4.* **Impact of Disabling G&A Heads Across Models.** Each value reports the change relative to the 0-head configuration. Evaluations are conducted on MMLU (Hendrycks et al., 2021), LAMBADA (Paperno et al., 2016), Grade School Math (GSM8K) (Cobbe et al., 2021), SWDE (Arora et al., 2024), Big-Bench Hard (BBH) (Suzgun et al., 2022), and knowledge-focused tasks defined in Section 3.1. Note that (1) GSM8K and BBH are evaluated using the "exact-match, flexible-extract" metric; (2) GSM8K is omitted for Llamba models, which were not trained on mathematical datasets and are not intended for arithmetic reasoning.

```
scallops:84
neuropsychiatry:67
The value of the key 'scallops' is
```

**Evaluation Setup.**    We use dictionaries containing key-value pairs and evaluate the model on randomly generated examples. For each example, we assess the model's ability to retrieve the correct value under two settings:

1. **Generation:** The model is prompted with the full dictionary and asked to generate 10 additional tokens. We check whether the correct value appears anywhere in the generated output.

2. **Answer Scoring:** Each possible value from the dictionary is appended to the end of the prompt, and the log-probability of the resulting sequence is computed. The value assigned the highest log-probability is treated as the model's prediction.

The key difference between these strategies lies in how the model expresses its beliefs. Generation allows the model to unfold a short sequence of reasoning steps, operating within a richer space of token-level interactions. However, this flexibility comes at the cost of interpretability, making it harder to surgically probe the model's internal mechanisms. Answer scoring, by contrast, restricts the model to a fixed set of candidates, offering a more direct readout of its internal ranking.

This distinction is important because generation can surface hallucinations—cases where the model outputs values not present in the dictionary or otherwise digresses from the task—while answer scoring is restricted to valid options. We find that the heads responsible for hallucinations differ from those that enable correct retrieval. This stands in contrast to the formulation of Wu et al. (2024), who used generation alone and concluded that "retrieval heads" may be responsible for hallucinations.

**Prompt Format.**    We test two variants of the query prompt that differ only in their final character:

1. `'The value of the key 'scallops' is'` and

2. `'The value of the key 'scallops' is '` (with a trailing space).

Though superficially similar, these variants elicit markedly different behaviors under Answer Scoring. In variant (1), the model must evaluate completions like `'The value of the key 'scallops' is84'`, which lacks a space and results in unnatural tokenization. Surprisingly, this unnatural format is highly diagnostic: head-level retrieval signals emerge clearly with as few as 20–30 key-value pairs. In contrast, variant (2), which uses the more natural space-delimited form, requires much longer dictionaries to produce similar head-level effects. For comparison, Wu et al. (2024) relied on 35K-token contexts to observe such patterns—well beyond typical context lengths.

We hypothesize that this discrepancy reflects a shift in the distribution of retrieval effort across the model. When the prompt ends cleanly on a token boundary, the model can offload the task to a small number of dedicated retrieval heads. By contrast, introducing a continuation token (like a space) appears to encourage more distributed processing across heads.

**Head-Ablation Protocol.**    To identify attention heads involved in retrieval, we systematically ablate each head by zeroing out its output projection and measure the resulting drop in accuracy under Answer Scoring (without the trailing space). This procedure isolates heads that are critical for retrieval while keeping computational overhead low.

Overall, this setup offers a clean and flexible diagnostic for in-context retrieval—independent of natural language or factual knowledge—and clarifies how individual heads contribute to this core algorithmic skill.

