# OpenReview forum: "Understanding the Skill Gap in Recurrent Language Models: The Role of the Gather-and-Aggregate Mechanism"
_ICML.cc/2025/Conference — ICML 2025 poster_

### Official Review · Reviewer_7vUD · 2025-03-09

**Overall Recommendation:** 4

**Summary:**

This paper seeks to investigate why SSMs underperform Transformers on on retrieval tasks. The paper identifies a "Gather-and-Aggregate"  (G&A) mechanism that emerges in Transformers and SSMs (though with some differences). The authors find that Transformers and SSMs concentrate this G&A mechanism in just a few heads and disabling them can significantly reduce MMLU scores while maintaining scores on knowledge intensive tasks, suggesting the importance of G&A for retrieval. Further experiments are performed to study the importance of these heads in SSM and hybrid models.

## Update after rebuttal: I maintain my score as my main questions have been addressed.

**Claims And Evidence:**

The claims in the paper are generally well supported with empirical evidence from a series of ablation studies. Pretrained LLama-3.1-8B, Llamba-9B and Falcon-Mamba-7B models are studied and ablated to establish consistency of findings across Transformers and SSMs.

The evidence that removing specific heads drastically impacts MMLU while having much less effect on "Knowledge" tasks provides evidence for the claims around the importance of these heads for retrieval. This is also supported with an additional experiment on a KV-retrieval task.

The experiments with the hybrid model also provides evidence for the paper's claim to shed light on why hybrid models can be effective.

Weaknesses of claims and evidence:
- Line 061 claims in the intro that the hybrid analysis "provides valuable insights into how to effectively place attention heads in a hybrid model to optimize performance", however I do not think this claim is well supported. In Section 6.4, the hybrid replacment experiments swap in individual attention layers from the Llama model to its ssm distilled Llamba. This experiment suggests layer 17 is important for MMLU. But this only again confirms attention layers are important, but does not say anything about optimal performance.
- The general claims could be strengthened with a broader set of evaluations than just MMLU and the kv retrieval task, e.g. summarization, long context Q&A, instruction following, etc.

**Essential References Not Discussed:**

The paper appropriately cites Jelassi et al. 2024 and Wen et al. 2024, for the difficulties that SSMs face in retrieval and copying, however there is a growing body of work surrounding this issue that would be useful to include citations to in order to provide an unfamiliar audience more theoretical and empirical context: Park et al. 2024 (https://arxiv.org/abs/2402.04248),  Arora et al 2023 (https://arxiv.org/abs/2312.04927), Arora et al 2024 (https://arxiv.org/abs/2402.18668), Blouir et al 2024 (https://arxiv.org/abs/2411.01030).

**Experimental Designs Or Analyses:**

Discussed above in Claims and Evidence section.

**Methods And Evaluation Criteria:**

The general methods and evaluations, in particular the layer and head ablation approaches, do make sense for analysis of where and how retrieval capabilities are implemented in these models.

**Other Comments Or Suggestions:**

- Line 140, right column, says that all three models "encode retrieval capabilities in the same way across layers". However the next sentence contradicts this. I assume the first sentence has a typo.
- Line 160 right column, Llama-9B should be Llamba-9B.
- Section 3.4 caveats says that while the models tested exhibit the same two-layer mechanism, the interaction does not always occur in consecutive layers. How do you know this? Based on what evidence if you didn't see this in your experiments?

**Other Strengths And Weaknesses:**

Strengths:
- Paper provides a compelling explanation for the retrieval gap between transformers and SSMs
- The identification of the G&A mechanism being concentrated in only a few heads is quite an interesting finding that could lead to deeper explorations in future works.
- The practical connection to hybrids and providing some evidence for why they can still work well is relevant to the community
- The visual representations of the G&A mechanism are useful

Weaknesses:
- While the paper does a decent job describing and qualitatively visualizing the G&A mechanism, the argument would be strengthened with more empirical measurements of how much gathering and aggregating is performed in different heads of different models.
- The paper would be strengthened by broadening the evaluations beyond just MMLU and kv-retrieval (discussed above). Could different types of task requiring retrieval require different mechanisms or different heads?
- The paper suggests its findings could inform better hybrid designs, but the current evidence for this in the paper is lacking (discussed more above).

**Questions For Authors:**

Most of my main concerns are discussed above.

1. Am I missing information you provide regarding "how to effectively place attention heads in a hybrid model to optimize performance"? If not, I would recommend rewording these claims. I suspect this is hard to show by just ablating models already pretrained, since it is hard to compare to the counterfactual of training a differently designed model (e.g. informed by the G&A findings) from scratch.
2. Have you tried other evaluations that require some form of retrieval, e.g. summarization, long context Q&A, etc?
3. All models explored here are around the 8B parameter level. Do larger models exhibit the same patterns?

**Relation To Broader Scientific Literature:**

The paper extends prior work on retrieval heads in Transformers and connects it to recent studies on SSMs, hybrids and the weaknesses of SSMs in retrieval intensive tasks.

**Theoretical Claims:**

The paper is primarily empirical.

---

> ### Author Rebuttal · Authors · 2025-04-01
>
> We’re glad the reviewer found our analysis of the Gather-and-Aggregate (G&A) mechanism compelling—both in its development within SSMs and its implications for hybrid models. Their main concern centers on the need for stronger empirical support and broader evaluations. We respond to each of these concerns below.
>
> > Section 3.4 caveats says that while the models tested exhibit the same two-layer mechanism, the interaction does not always occur in consecutive layers. How do you know this? Based on what evidence if you didn't see this in your experiments?
>
> Our experiments happen to observe G&A in consecutive layers, but we do not claim this is a universal pattern. For example, [1] shows that retrieval mechanisms can span non-consecutive layers. Thank you for the question — we’ve updated the paper to clarify this point.
>
> > Have you tried other evaluations that require some form of retrieval, e.g. summarization, long context Q&A, etc?
>
> We agree that broader evaluations are essential to distinguish general retrieval from format-specific processing. In response to reviewers’ suggestions, we added experiments on tasks with diverse structures and retrieval demands:
>
> |Task|score (%)|G&A masked|
> |-|-|-|
> |gsm8k_cot|33.6|No|
> |gsm8k_cot| 5.8|Yes|
> |gsm8k|40.3|No|
> |gsm8k|10.2|Yes|
> |ifeval|31.1| No|
> |ifeval|21.3|Yes|
>
> As shown, masking just 8 G&A attention heads across two layers in Zamba2-7B causes substantial drops in performance, reinforcing that G&A supports retrieval across a range of tasks.
>
> We are also expanding our evaluation suite to include tasks such as LongBench (long-context summarization, QA, and code), RULER (needle-in-a-haystack and variable tracking), and broader chain-of-thought (CoT) benchmarks. While we were unable to include these results during the rebuttal phase due to time constraints, we are actively working on them in response to your suggestion.
>
> > While the paper does a decent job describing and qualitatively visualizing the G&A mechanism, the argument would be strengthened with more empirical measurements of how much gathering and aggregating is performed in different heads of different models.
>
> We agree that quantifying the presence of G&A across heads and models is an important direction that would strengthen the paper. This kind of analysis would also directly inform practical questions, such as the one the reviewer raised about where to place attention heads in hybrid models.
>
> So far, we have identified G&A heads through manual inspection, which is time-intensive and does not yet scale easily across models. However, as noted in our response below on hybrid models, findings from [2] provide complementary evidence on the emergence and positioning of such heads in Transformers, further underscoring the relevance of this line of investigation.
>
> We see this as a promising direction for future work, including developing automated tools to more systematically measure G&A across architectures.
>
> > All models explored here are around the 8B parameter level. Do larger models exhibit the same patterns?
>
> We acknowledge this as a limitation and agree that evaluating larger models would help validate the broader applicability of our findings. However, the next scale of available models is typically 70B+, which exceeds our academic compute resources and would require substantial engineering effort. That said, similar heads have been observed in Chinchilla-70B by [1], as well as in other works (see “The sample of models is too limited” in our response to reviewer BU3Q), suggesting that these mechanisms likely scale to larger models.
>
> > Am I missing information you provide regarding "how to effectively place attention heads in a hybrid model to optimize performance"?
>
> Thank you for raising this. We now clarify it more explicitly in the revised paper. Our approach to hybrid design involves:
> 1. Hybrid Replacement: Given a pretrained pure SSM model, we identify prominent G&A instances (Section 6.4). These locations signal where SSM heads are replaced with attention heads, improving retrieval and narrowing the performance gap.
> 2. Placement Guidance: G&A consistently arises in middle layers—for example, layers 16–17 (of 32) in LLaMA, 35–36 (of 64) in Falcon-Mamba, and 47 and 59 (of 80) in Zamba-7B. This pattern is consistent with prior findings [2], where such attention heads emerge in middle layers. Placing attention heads at similar depths complements the SSM backbone when global context is most needed.
>
> **References:**
>
> [1] Lieberum et al., 2023 — arxiv.org/abs/2307.09458
>
> [2] Zheng et al., 2024 — arxiv.org/abs/2409.03752

---

### Official Review · Reviewer_vZz2 · 2025-03-12

**Overall Recommendation:** 4

**Summary:**

I am not an expert in transformers and SSMs.
The authors reverse engineer language models to show that retrieval capabilities are supported by distinct parts of the networks compared to overall knowledge.
By a systematic lesioning of layers, they identify that (at least) two layers are needed to support retrieval. The first of the layers contains heads which aggregate information from a segment and encode it in the final token. The second layer contains heads which process these tokens to decide on the correct response. The smooth nature of SSMs makes it harder for them to constrain activation from neighboring tokens, and thus they struggle more with these tasks.

**Claims And Evidence:**

Yes. The lesion experiments show that two layers are needed, and that they operate in the described manner on MMLU.

**Essential References Not Discussed:**

not aware

**Experimental Designs Or Analyses:**

Not in detail

**Methods And Evaluation Criteria:**

Yes. The benchmarks are designed to tease apart knowledge from retrieval.

**Other Comments Or Suggestions:**

None.

**Other Strengths And Weaknesses:**

This is a nice systematic analysis of the underlying mechanisms of language models. The separation to knowledge and retrieval, and the identification of a small number of elements responsible for retrieval is an important finding that helps understand and improve models.

**Questions For Authors:**

None.

**Relation To Broader Scientific Literature:**

Not an expert in this literature.

**Theoretical Claims:**

Not relevant

---

> ### Author Rebuttal · Authors · 2025-04-01
>
> We thank the reviewer for their positive remarks. We’re glad the conceptual separation between knowledge and retrieval, along with the identification of the elements driving retrieval, was found to be valuable.
> If any questions arise, we would be happy to address them.

---

### Official Review · Reviewer_BU3Q · 2025-03-13

**Overall Recommendation:** 3

**Summary:**

This paper investigates the performance gap between Transformer and State-Space Model (SSM) language models, focusing on retrieval capabilities. The authors identify a "Gather-and-Aggregate" (G&A) mechanism that emerges in both architectures but is implemented more effectively in Transformers. This two-part mechanism consists of a "Gather Head" that condenses segment information into the last token, and an "Aggregate Head" that processes these condensed representations. The key finding is that despite different architectures, both models develop similar mechanisms, but SSMs struggle with the Aggregate component due to their fixed state size. Remarkably, these capabilities are concentrated in just a few critical heads, and disabling a single head can severely degrade performance on tasks like MMLU (dropping from 66% to 25%). The paper also shows that hybrid models naturally assign Aggregate heads to attention layers, explaining their success.

**Claims And Evidence:**

The authors present a methodical investigation through:

- Ablation studies to demonstrate the critical role of specific layers and heads
- Visualization of attention patterns in both Transformer and SSM models
- Performance measurements on MMLU and knowledge-focused tasks
- Analysis of hybrid models showing where Aggregate heads are allocated

**Essential References Not Discussed:**

Meng et al. (2022) "Locating and Editing Factual Associations in GPT" discusses similar mechanisms for fact storage and retrieval

**Experimental Designs Or Analyses:**

Several experimental weaknesses undermine the strength of the conclusions:

1. The sample of models is too limited (three models) to support broad claims about architectural differences.
2. The paper focuses heavily on MMLU and simplistic KV-Retrieval tasks without testing on a diverse range of retrieval-focused benchmarks.
3. The hybrid replacement experiment is clever but confounded by potential interactions between layers that aren't accounted for.

**Methods And Evaluation Criteria:**

The evaluation approach has significant limitations. While the authors distinguish between knowledge tasks and retrieval-heavy tasks, this binary categorization oversimplifies the complex capabilities required for different benchmarks. MMLU is treated as primarily a retrieval task, but prior work has demonstrated it tests a broader range of skills.

The ablation methodology is appropriate but limited in scope. While identifying critical components through knockouts is sound, the paper fails to systematically explore alternative hypotheses or control for confounding factors. For example, the authors don't adequately address whether the effect is truly due to the proposed mechanism or other properties of the affected heads.

The KV-Retrieval task provides a cleaner test case, but the artificial nature of this task limits generalization to real-world model performance differences.

**Other Comments Or Suggestions:**

The paper would be substantially strengthened by:
- Testing on a broader range of models and tasks
- More rigorously controlling for alternative explanations
- Addressing the limitations and generalizability of the findings

**Other Strengths And Weaknesses:**

Strengths:
- The paper introduces a novel perspective on architectural differences
- The visualization of head activations effectively illustrates the proposed mechanism
- The hybrid model analysis provides useful insights for architectural design

Weaknesses:
- The paper focuses heavily on MMLU as a proxy for retrieval capabilities. A wider range of retrieval-focused benchmarks would strengthen the generalizability
- There is limited exploration of how the G&A mechanism might interact with other architectural components (like normalization layers)
- The authors don't fully explore how the G&A mechanism might be improved in SSMs

**Questions For Authors:**

- Is G&A mechanism truly the causal factor in performance differences rather than a correlate of some other more fundamental architectural difference? Additional control experiments would be necessary to establish causality.
- Have you explored whether the same performance gap exists for tasks that don't follow the multiple-choice format of MMLU? This would help distinguish between general retrieval capabilities versus format-specific processing.
- What evidence do you have that the G&A limitation is fundamental to SSM architecture rather than an artifact of current implementations or training methods?

**Relation To Broader Scientific Literature:**

The authors acknowledge key related work including:

- Olsson et al. (2022) and Elhage et al. (2021) on induction heads
- Lieberum et al. (2023) on "Content Gatherer Head" and "Correct Letter Head"
- Wu et al. (2024) on retrieval heads

**Theoretical Claims:**

The paper does not have a theoretical proof.

---

> ### Author Rebuttal · Authors · 2025-04-01
>
> We appreciate the reviewer’s recognition of the novelty in our approach to architectural differences and hybrid experiments. Their comments raise valuable questions about performance gaps, generality across models and tasks, and the role of other components. We address each point below.
>
>
> > There is limited exploration of how the G&A mechanism might interact with other architectural components (like normalization layers)
>
> We agree this is a valuable direction. Our focus is on understanding retrieval mechanisms in established model architectures, where normalization practices are typically standardized. While these interactions interest us, we leave their exploration to future work.
>
>
> > The authors don't fully explore how the G&A mechanism might be improved in SSMs
>
> While improving the G&A mechanism in SSMs is valuable, prior work has shown that retrieval limitations are inherent to all RNN-based models due to fixed-size memory, which constrains tasks like KV-Retrieval [9–11]. We therefore focused on hybrid models that directly address this issue, showing how they mitigate memory constraints rather than pursuing limited incremental gains within pure SSMs.
>
> >Is G&A mechanism truly the causal factor in performance differences rather than a correlate of some other more fundamental architectural difference?
>
> Thank you for raising this—we have clarified this in the revised paper as well.
> To support a causal role for G&A in MMLU performance differences between architectures, we present the following:
> 1. MMLU is bottlenecked by retrieval, not knowledge (Sections 3 & 5): Models can do well on MMLU despite weak general performance, but fail without a working G&A mechanism to retrieve letters from the prompt.
> 2. G&A enables retrieval in language models (Sections 4 & 5): the same G&A heads drive synthetic KV retrieval performance.
> 3. SSMs struggle with G&A (Section 6) due to their fixed memory structure [9–11].
>
> This forms a causal chain: architectural constraints → impaired G&A → retrieval failure → MMLU drop. To further support this interpretation, we added prior work [7–9] linking retrieval—and thus G&A—to the Transformer–SSM gap.
>
> If any part of this causal chain remains unconvincing, we’d appreciate clarification on which link is in doubt, so we can design targeted control experiments to isolate it.
>
> > What evidence do you have that the G&A limitation is fundamental to SSM architecture rather than an artifact of current implementations or training methods?
>
> It is well-established, both theoretically and empirically, that SSMs are weaker at retrieval than attention-based models due to their fixed-size memory [9–11]. A key takeaway of our work is that the G&A mechanism, particularly the Aggregate head, drives retrieval. Given this, it is natural to expect that limitations in G&A may be fundamental to the SSM architecture.
>
> Our masking experiment (Section 4.2) further shows that SSM-based G&A fails to implement the mechanism correctly, as recurrent dynamics smooth attention across tokens. This supports the view that the limitation stems from architectural properties, rather than optimization.
>
> > The sample of models is too limited.
>
> We agree that evaluating more models would strengthen the paper.  We chose the two available SSMs (Mamba-1, Mamba-2) and a Transformer to ensure architectural diversity, given the manual and resource-heavy process of identifying relevant heads.
> In response to this suggestion, we are running tests on the entire Falcon and LLaMA 3 families. If there are other models the reviewers would like to see included, we would be glad to add them.
>
> That said, similar retrieval behaviors have been observed in LLaMA-2 [2–6], Chinchilla [1], Phi-2 [4, 6], Falcon [4], Mixtral-8x7B [4], and Yi [6], ranging from 7B to 70B, suggesting that our findings are representative.
>
> > Have you explored whether the same performance gap exists for tasks that don't follow the multiple-choice format of MMLU?
>
> Due to space limits, we address this under 7vUD (‘other evaluations’). TLDR: gsm8k and ifeval confirm the same G&A head dependency; we’re expanding to diverse tasks.
>
>
> > The hybrid replacement experiment is clever but confounded by potential interactions between layers that aren't accounted for.
>
> We agree that the head’s effect could be more directly tested. To address this, we replaced only the Aggregate head—leaving the rest of the layer unchanged—and observed a similar 17-point gain in MMLU. Thank you for pointing this out.
>
> **References:**
>
> [1] arxiv.org/abs/2307.0945
>
> [2] arxiv.org/abs/2404.15574
>
> [3] arxiv.org/abs/2409.01659
>
> [4] arxiv.org/abs/2402.12483
>
> [5] arxiv.org/abs/2407.15018
>
> [6] arxiv.org/abs/2402.01781
>
> [7] arxiv.org/abs/2312.04927
>
> [8] arxiv.org/abs/2402.18510
>
> [9] arxiv.org/abs/2406.07887
>
> [10] arxiv.org/abs/2402.01032
>
> [11] arxiv.org/abs/2501.00658

---

> > ### Comment · Reviewer_BU3Q · 2025-04-09
> >
> > I thank the authors for a detailed rebuttal. I think the authors have addressed most of my concerns, and I decide to raise my score.

---

### Official Review · Reviewer_x3LE · 2025-03-23

**Overall Recommendation:** 3

**Summary:**

**Updates after rebuttal: I increased my score in light of authors' rebuttal (particularly on the distillation procedure)**

*I appreciate the authors efforts in explaining and demonstrating how their distillation recipe can be used to distill hybrid models from scratch, motivated from the Gather-and-Aggregate mechanism. This addresses my main concern. I have updated my score to 3.*

---

This paper investigates the Gather-and-Aggregate (G\&A), a mechanism for solving retrieval tasks that emerges in both SSMs and transformers. The authors found that only a small number of such G&A heads exist in pretrained language models. From empirical results: the authors showed the gap between SSMs and transformers on retrieval is mainly due to these small number of G\&A heads; the authors also showed SSMs struggle to implement the Aggregation head, whereas the hybrid models can mitigate this issue.

**Claims And Evidence:**

1. The authors claimed that "pretrained language models contain only a small number of Gather-and-Aggregate head", but only empirically evaluate this claim on three pretrained language models (Llama-3.1-8B, Lambda-9B, Falcon-Mamba-7B). I suggest the authors either weaken their claim or support it with stronger evidence across many more language models.

2. The authors claimed that Gather-and-Aggregate mechanism is an extended version of the mechanism found in Lieberum et al. (2023) in the introduction, see Line 019-029. But from the body of the paper, it is suggested that the Gather and Aggregate head play the same role as the Content-Gatherer head and Correct-Letter head in Lieberum et al. If so, why the new name, and is the extension lies in investigating such mechanism in SSMs (beyond transformers done in Lieberum et al.)? If not, what are the differences among the definitions of these heads?

**Essential References Not Discussed:**

The following is another recent Hybrid model, which came out around the same time as MOHAWK. The authors should discuss their choice of Hybrid model and whether such choice affects the findings.
[1] Wang, Junxiong, et al. "The mamba in the llama: Distilling and accelerating hybrid models." Advances in Neural Information Processing Systems 37 (2024): 62432-62457.

The following line of works are theoretical papers proving the retrieval mechanisms implemented in transformer for induction head task:
[2] Bietti, Alberto, et al. "Birth of a transformer: A memory viewpoint." Advances in Neural Information Processing Systems 36 (2023): 1560-1588.
[3] Sanford, Clayton, Daniel Hsu, and Matus Telgarsky. "Transformers, parallel computation, and logarithmic depth." Forty-first International Conference on Machine Learning.

**Experimental Designs Or Analyses:**

1. In Section 6.3, the authors disabled all potential Aggregate heads across all attention layers, rather than disabling only the identified Aggregate head, which led to an decline of around 4x of performance in MMLU. However, what if there are other mechanisms implementing in the attention layers that contribute to the MMLU task (which also get disabled in this process)?

**Methods And Evaluation Criteria:**

1. Most methods proposed in this work to mechanistically interpret the Gather-and-Aggregate mechanism are appropriate. However, the method in Section 4.3 requires further evidence. Specifically, "Evaluating the MMLU benchmark with these masks showed no loss in performance" -- which loss of performance baseline do the authors refer to? It will be nice to have a table showing the original MMLU performance, the performance after disabling all heads except the G\&A heads, and the performance after disabling all heads except the G\&A heads with attention mask.

**Other Comments Or Suggestions:**

None

**Other Strengths And Weaknesses:**

Strengths: The paper is well-written overall. I find the idea of identifying the crucial heads in language models for MMLU or general retrieval tasks interesting. I also appreciate the application of such idea in the hybrid models.

Weaknesses:
1. The G\&A mechanism is proposed in the previous work by Lieberum et al. (2023) focusing on transformers. The current work does not seem to offer additional mechanistic interpretability insights on such mechanism.

2. It is well known empirically and theoretically that SSM struggles at retrieval compared to transformers (Jelassi et al., [3] Sanford et al.). Thus, it also seems quite natural that SSM is worse at implementing the Aggregate head.

3. The investigation of the hybrid model is interesting. Nonetheless, it will be much more convincing if the authors show positive results of distilling a transformer to a hybrid model by keeping the Aggregation head intact in the attention layer.

4. Experiments: more details on Sec 4-6 should be supplemented.

**Questions For Authors:**

See questions in the previous sections, indexed as:
- Claims and Evidence: C1 - C2;
- Methods and Evaluation: M1
- Experimental Design: E1

**Relation To Broader Scientific Literature:**

This work is broadly related to the mechanistic interpretability of language models. It builds on the earlier work by  Lieberum et al. (2023) on identifying the Content-Gatherer and Correct-Letter head mechanisms in transformers for solving the MMLU task. It examines how such mechanisms arise in other transformer-based language models, SSMs, and hybrid models.

**Theoretical Claims:**

None

---

> ### Author Rebuttal · Authors · 2025-04-01
>
> We appreciate the reviewer’s positive feedback on our approach to identifying retrieval heads in language models and their relevance to hybrid architectures. Below, we address the reviewer’s concerns about terminology, empirical support, and methodological clarity.
>
> > It is well known empirically and theoretically that SSM struggles at retrieval compared to transformers. Thus, it also seems quite natural that SSM is worse at implementing the Aggregate head.
>
> We agree that this limitation seems natural. Our work sharpens this intuition by bridging the theoretical limitations observed in simplified SSM variants with the retrieval failures seen in full-scale language models. We show that the known weaknesses in the former manifest in the latter specifically through G&A instances. While this perspective may seem natural in retrospect, the behavior of Aggregate heads—and the mechanisms enabling them—had not been clearly understood in this setting.
>
> Surprisingly, we find that SSM-based G&A mechanisms fail even on simple short-context tasks—like retrieving a letter in MMLU—suggesting they fall short well before hitting their theoretical limit.
>
> > From the body of the paper, it is suggested that the Gather and Aggregate head play the same role as the Content-Gatherer head and Correct-Letter head in Lieberum et al. If so, why the new name, and is the extension lies in investigating such mechanism in SSMs (beyond transformers done in Lieberum et al.)? If not, what are the differences among the definitions of these heads?
>
> Thank you for raising this point. We will update the paper to clarify this.
> Our contribution lies not in proposing new head types, but in refining the understanding of their roles and how they support retrieval. Our core finding is that in both Transformers and SSMs, retrieval emerges not from a single head, but from a coordinated mechanism involving two roles: one head gathers the location of the target, and another aggregates the content.
>
> This framing extends prior work in three ways:
> 1. “Correct Letter” heads (Lieberum et al.) do more than select the correct option in multiple-choice tasks — they exhibit broader retrieval behavior (Sections 4.2 & 5),
> 2. Retrieval relies on coordinated interaction between heads, rather than a single head as suggested in Wu et al. (Sections 3 & 4),
> 3. We analyze this mechanism in SSMs, which lack attention and have not been studied in this context.
>
> The new terminology reflects these empirical findings rather than renaming existing concepts. We also note that we retain “Gather” heads as shorthand for the “Content-Gatherer” heads described in Lieberum et al.
>
> > The authors claimed that "pretrained language models contain only a small number of Gather-and-Aggregate head", but only empirically evaluate this claim on three pretrained language models.
>
> Due to space limits, we address this under BU3Q (‘sample of models is too limited’). TLDR: we are expanding our evaluation to include additional models.
>
> > Section 4.3: It will be nice to have a table showing the original MMLU performance, the performance after disabling all heads except the G&A heads, and the performance after disabling all heads except the G&A heads with attention mask.
>
> Thank you for pointing this out. We’ve added the table to Section 4.3 as suggested.
>
> > The investigation of the hybrid model is interesting. Nonetheless, it will be much more convincing if the authors show positive results of distilling a transformer to a hybrid model by keeping the Aggregation head intact in the attention layer.
>
> Thank you for the suggestion. We address this point in the “Hybrid Replacements” experiment (Section 6.4), where we systematically replaced each layer of the distilled Llama-8B model with its counterpart from Llama-3.1-8B and evaluated the effect on MMLU (without fine-tuning). As shown in Figure 7, most replacements had little or negative impact, but substituting Layer 17—associated with a strong Aggregate head (Table 1)—led to a clear gain, improving performance from 33% to 50%.
>
> > In Section 6.3 ... what if there are other mechanisms implementing in the attention layers that contribute to the MMLU task?
>
> We agree with the reviewer’s point and have refined our intervention to better isolate the role of the G&A mechanism in response. Specifically:
> 1. Head Selection: We restricted masking to a small set of manually identified G&A heads—L47H{17, 18, 25} and L59H{17, 21}.
> 2. Token Selection: Masking was applied only to specific tokens in the final attention row involved in the G&A pattern (e.g., “:” attending to choices “A–D”).
>
> With knowledge-task accuracy stable at 70% and MMLU dropping from 64% to 35%, this targeted intervention indicates that other attention patterns contributing to MMLU were not impacted.
>
> **References**:
>
> [1] Lieberum et al., 2023 — arxiv.org/abs/2307.09458
>
> [2] Wu et al., 2024 — arxiv.org/abs/2404.15574

---

> > ### Comment · Reviewer_x3LE · 2025-04-03
> >
> > I thank the authors for the detailed response. Follow-up on the hybrid model design: The strategies and evidences in the submitted version on improving hybrid models is a *post-hoc* adjustment: take a pretrained hybrid model, and replace the distilled SSM layer with the corresponding crucial Gatherer attention layer. My question, similar to Reviewer 7vUD, lies in whether this offers insights in distillation *from scratch*. The authors rebuttal to 7vUD discussed more details on their hybrid model design  1. Hybrid Replacement and 2. Placement Guidance. But is this proposed design supported by empirical evidence?

---

> > > ### Author Response · Authors · 2025-04-05
> > >
> > > To answer this, we first formalize the distillation process. Given a teacher model $T$ and a target architecture $A$, a distillation algorithm $D$ produces a student model $D(T, A)$ that imitates $T$ using architecture $A$.
> > >
> > > We believe that the reviewer is asking twofold questions:
> > > 1. Instead of running $D(\text{Llama}, S)$ where $S$ is a pure SSM, and then swapping in attention layers, is it possible to define a hybrid architecture $H$ and directly run $D(\text{Llama}, H)$?
> > > 2. Is the proposed design supported by empirical evidence?
> > >
> > > If we’ve misunderstood or if you’d like us to expand further, we’re happy to clarify.
> > >
> > > To answer this, we consider the MOHAWK framework [1], which has three steps. The two relevant ones are:
> > >
> > > - **$D = \text{MOHAWK}_2$ (Layer-to-Layer Distillation):** Each student layer independently mimics the corresponding teacher layer by minimizing the L2 norm of their output.
> > > - **$D = \text{MOHAWK}_3$ (Knowledge Distillation):** The Step 2 model is further fine-tuned end-to-end with a cross-entropy objective over logits, using a small data subset. This effectively encapsulates $\text{MOHAWK}_2$ with further distillation.
> > >
> > > To help clarify our discussion, we note that the Hybrid Replacement experiment uses only $\text{MOHAWK}_2$, and our understanding is that “distillation from scratch” refers to step 3.
> > >
> > > > **Can we define a hybrid architecture $H$ and directly run $D(T, H)$?**
> > >
> > > Yes. Because $MOHAWK_2$ distills each layer independently, swapping in an attention layer *after* distillation is equivalent to defining a hybrid architecture $H_i$ *before* distillation, where $H_i$ uses attention at layer $i$ and SSMs elsewhere. In other words, running $D(T, S)$ and then swapping layer $i$ yields the same result as running $D(T, H_i)$. **This gives us a principled way to define a hybrid model upfront, informed by the teacher’s architecture**.
> > >
> > > > **Is the proposed design supported by empirical evidence?**
> > >
> > > We believe our ablations in Section 6.4 help guide the design of an effective hybrid. Figure 7 shows that across all $i$, $\text{MOHAWK}_2(T, H_i)$ performs best when attention is placed at $i = 17$, the same layer as the teacher’s Aggregate head.
> > >
> > > Formally, for all $i \neq 17$, we have $MOHAWK_2(Llama, H_i) < MOHAWK_2(Llama, H_{17})$, indicating that this alignment yields the strongest result.
> > >
> > > This experiment demonstrates:
> > > - How to identify an effective hybrid architecture $H_i$ using the teacher’s structure.
> > > - That $\text{MOHAWK}_2(T, H_i)$ significantly outperforms $\text{MOHAWK}_2(T, \text{PureSSM})$—e.g., on MMLU, improving from 32.3 to 49.8.
> > >
> > > While $MOHAWK_3$ likely yields even stronger models, it is computationally more intensive since it distills layers jointly. Thus, we did not compute $MOHAWK_3(T, H_i)$ for all $i$. However, we evaluated it for the best-performing hybrid (layer $i = 17$) and found: $MOHAWK_3(T, H_{17})$ improves MMLU to **62%** using only 3,000 optimization steps, whereas Llamba-8B ran $MOHAWK_3(T, \text{PureSSM})$ for tens of thousands of steps and achieved an MMLU score of 61%
> > >
> > > |Benchmark|Step2|Step2 (keepingL17)|Step2+3 (keepingL17)|PureSSM Step2+3 (Llamba-8B)|LLaMA-3.1-8B|
> > > |-|-|-|-|-|-|
> > > |Knowledge Tasks |64.1|64.0|68.6|68.7|69.0|
> > > |**MMLU**|**32.3**|**49.8**|**62.0**|**61.0**|**68.0**|
> > >
> > > Where “Knowledge Tasks” refers to the average of ARC-Challenge, ARC-Easy, PIQA, Winogrande, and OpenBookQA (Section 3).
> > >
> > > These results validate that our method provides a practical recipe for hybrid distillation from scratch:
> > >
> > > 1. **Identify** the G&A heads in the teacher $T$.
> > > 2. **Define** a hybrid student $H$: mostly SSMs, with attention in the layers where strong Aggregate heads appear (e.g., layer 17 in LLaMA 3.1).
> > > 3. **Run** a distillation algorithm $D(T, H)$. We’ve shown that both $\text{MOHAWK}_2$ and $\text{MOHAWK}_3$ yield stronger hybrids than the pure SSM baseline (e.g. $Llamba=D(Llama, S)$)
> > >
> > > We're happy to revise further if we’ve misunderstood or if more detail would help. Otherwise, we’ll revise the paper to be clearer in light of your questions.
> > >
> > > References:
> > >
> > > [1] arxiv.org/abs/2408.10189

---

### Decision · Program_Chairs · 2025-05-01

**Decision:**

Accept (poster)

**Comment:**

The paper provides a comparison of attention vs recurrent language models in a specific retrieval task that seems crucial for good performance on MMLU, namely a gather-and-aggregate mechanism that in particular helps with selecting the correct answer in multiple-choice questions, following a study of Lieberum et al.

The findings about the importance of this mechanism for MMLU performance, as well as how this contributes to the gap between SSMs and attention, provides compelling real-world evidence for the known issues of SSMs with retrieval tasks, which were previously studied mainly on toy tasks/models.

The reviewers mostly aligned towards accepting the paper. Please revise the paper to incorporate the clarifications that you provided to reviewers during the rebuttal, particularly regarding the hybrid replacement approach raised by reviewer x3LE and the causal role of your experimental interventions on the attention-vs-ssm gap.